# Entropic Mirror Monte Carlo

**Anas Cherradi** [1 2] **Yazid Janati** [3] **Alain Durmus** [4] **Sylvain Le Corff** [1] **Yohan Petetin** [5] **Julien Stoehr** [6 7]

## Abstract

Importance sampling is a Monte Carlo method which designs estimators of expectations under a target distribution using weighted samples from a proposal distribution. When the target distribution is complex, such as multimodal distributions in high-dimensional spaces, the efficiency of importance sampling critically depends on the choice of the proposal distribution. In this paper, we propose a novel adaptive scheme for the construction of efficient proposal distributions. Our algorithm promotes efficient exploration of the target distribution by combining global sampling mechanisms with a delayed weighting procedure. The proposed weighting mechanism plays a key role by enabling rapid resampling in regions where the proposal distribution is poorly adapted to the target. Our sampling algorithm is shown to be geometrically convergent under mild assumptions and is illustrated through various numerical experiments.

## 1. Introduction

Let $\pi$ be a probability distribution on $\mathcal{X} = \mathbb{R}^d$, known up to a multiplicative constant. Our objective is to estimate expectations of the form $\pi(f) = \int f(x)\pi(\mathrm{d}x)$ for measurable test functions $f$, in settings where direct sampling from $\pi$ is infeasible. Such situations naturally arise in Bayesian inference, rare event simulation, and high-dimensional statistical modeling.

A standard methodology for addressing this problem is importance sampling. Given a proposal distribution $\mu$ that dominates $\pi$, expectations under $\pi$ can be approximated by reweighting samples drawn from $\mu$ (Akyildiz & Míguez, 2021; Agapiou et al., 2017; Thin et al., 2021). The statistical efficiency of importance sampling critically depends on the mismatch between $\mu$ and $\pi$. In particular, it is well known that the mean squared error of self-normalized importance sampling estimators can grow exponentially with divergence measures such as the Kullback–Leibler or Rényi divergences between $\pi$ and $\mu$ (Agapiou et al., 2017; Chatterjee & Diaconis, 2018). This phenomenon severely limits the applicability of importance sampling in high-dimensional settings or when $\pi$ exhibits strong multimodality. In Grenioux et al. (2025), the authors highlight in particular that addressing multi-modality is a major challenge of sampling algorithms and discuss systematic evaluation of various samplers.

Adaptive importance sampling methods aim to mitigate this issue by iteratively updating the proposal distribution using weighted samples from previous iterations (Oh & Berger, 1993; Cappé et al., 2004; Cornuet et al., 2012). These approaches typically rely on parametric or non-parametric families of distributions and on variational criteria such as the minimization of the Kullback–Leibler divergence. Despite significant progress, designing adaptive proposals that simultaneously scale with the dimension and efficiently explore multimodal target distributions remains challenging.

Markov chain Monte Carlo methods provide an alternative strategy by constructing a Markov chain that admits $\pi$ as invariant distribution. These methods avoid the need to evaluate the normalizing constant of $\pi$ and enjoy strong asymptotic guarantees (Roberts & Rosenthal, 2004). However, in practice, their performance is often hindered by poor mixing properties, especially in multimodal settings where transitions between distant regions of high probability mass are unlikely. Moreover, while Markov chain Monte Carlo algorithms generate asymptotically exact samples, they do not directly yield tractable proposal distributions that can be evaluated pointwise, which prevents their direct use within importance sampling frameworks.

A promising direction to overcome these limitations consists in constructing sequences of probability measures $\{\mu_t\}_{t \in \mathbb{N}}$

---

[1]Sorbonne Université, Université Paris Cité, CNRS, Laboratoire de Probabilités, Statistique et Modélisation, LPSM, 75005 Paris, France [2]EMINES, University Mohammed VI Polytechnic, Benguerir, Maroc [3]Institute of Foundation Models, MBZUAI, 75002 Paris, France [4]Centre Mathématiques Appliquées, Institut Polytechnique de Paris, 91120 Palaiseau, France [5]Telecom SudParis, CNRS, Institut Polytechnique de Paris, 91011 Evry, France [6]CEREMADE, Université Paris-Dauphine, Université PSL, CNRS, 75016 Paris, France [7]Université Paris-Saclay, INRAE, AgroParisTech, UMR MIA Paris-Saclay, 91120 Palaiseau, France. Correspondence to: Anas Cherradi <anas.cherradi@emines.um6p.ma>.

*Proceedings of the 43rd International Conference on Machine Learning*, Seoul, South Korea. PMLR 306, 2026. Copyright 2026 by the author(s).

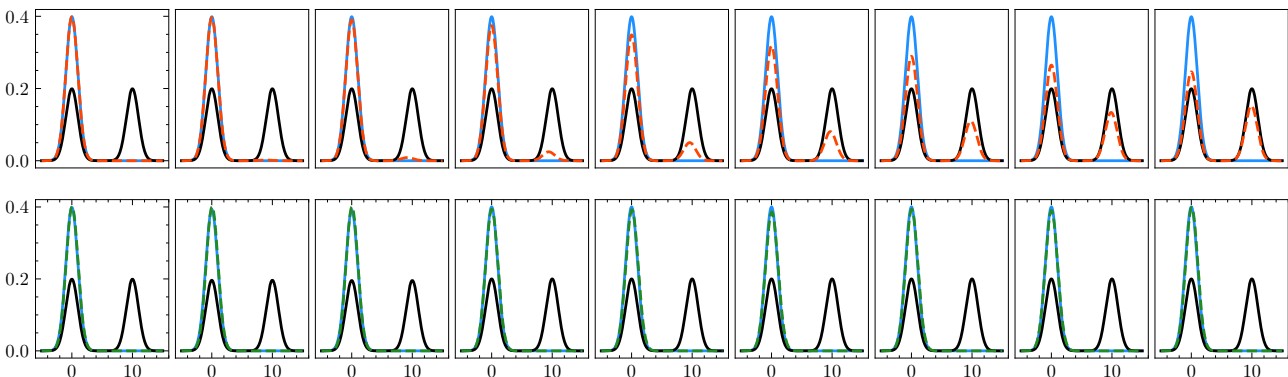

*Figure 1.* Intermediate sequence of proposal distributions build with the Entropic Mirror Descent for sampling from the bimodal target distribution $\pi = 0.5 \cdot \mathcal{N}(0,1) + 0.5 \cdot \mathcal{N}(10,1)$ (black solid line) and starting with distribution $\mu_0$ (blue solid line). Top row: theoretical sequence $\{\mu_t\}_{t \geq 0}$ (red dashed line). Bottom row: updates (green dashed line) when using a bimodal variational distributions build upon $5,000$ samples from $\{\mu_t\}_{t \geq 0}$.

that progressively approach the target distribution while enjoying explicit contraction properties. Recent works have shown that Entropic Mirror Descent maps provide a principled way to define such sequences, ensuring a geometric decrease of $\mathsf{KL}(\pi \parallel \mu_t)$ under suitable conditions (Beck & Teboulle, 2003; Dai et al., 2016; Korba & Portier, 2022; Bianchi et al., 2024). However, the resulting updates are often intractable and difficult to approximate in multimodal settings, as naive Monte Carlo approximations may fail to discover regions of high probability mass (see Figure 1).

In this paper, we propose *Entropic Mirror Monte Carlo* (EM2C) a new adaptive importance sampling methodology that augments Entropic Mirror Descent with Markovian dynamics to promote exploration while preserving contraction properties. The resulting mapping combines weighted samples drawn from the current proposal distribution with samples propagated through a Markov transition kernel. This construction yields a sequence of idealized updates that converges geometrically fast to the target distribution under mild assumptions. We further derive a practical stochastic implementation based on sampling, weighting and resampling steps, and demonstrate through numerical experiments that the proposed method efficiently handles challenging multimodal targets, including in high-dimensional settings.

The main contributions of this work can be summarized as follows.

- We propose a sequence of idealized updates for the proposal distributions that converges geometrically fast to the target distribution under mild assumptions. Each update is formulated as a mixture of two components: the first preserves the contraction properties of Entropic Mirror Descent and exploits regions already well represented by the current proposal, while the second leverages the exploratory capabilities of a Markov kernel.

- We show that, although the Unadjusted Langevin kernel induces a bias with respect to its invariant distribution, this bias remains controlled and does not hinder convergence, while the contraction effect of the Entropic Mirror Descent component is preserved.

- We introduce a novel importance sampling scheme to approximate the idealized algorithm and demonstrate its effectiveness through numerical experiments in various settings.

The paper is organized as follows. In Section 2, we present how to construct a sequence of probability measures that converges to the target distribution and enjoys explicit contraction guarantee and detail Entropic Mirror Descent. In Section 3, we introduce our stochastic implementation of Entropic Mirror Monte Carlo in Algorithm 1. The empirical performance of the algorithm is provided in Section 4.

## 2. Entropic Mirror Descent

**Notations and conventions.** Throughout the paper, the state space is $\mathcal{X} = \mathbb{R}^d$ and the reference dominating measure is the Lebesgue measure $\mathrm{Leb}$. We denote by $\mathbb{M}_1(\mathbb{R}^d)$ the set of probability measures on $\mathbb{R}^d$ and by

$$\mathbb{M}_\pi = \left\{ \mu \in \mathbb{M}_1(\mathbb{R}^d) : \pi \gg \mu \text{ and } \mu \gg \pi \right\}$$

the set of probability measures that are equivalent to $\pi$.

We do not distinguish probability measures from their densities with respect to $\mathrm{Leb}$. Accordingly, when $\mu \ll \mathrm{Leb}$, we write $\mu(\mathrm{d}x) = \mu(x)\,\mathrm{Leb}(\mathrm{d}x)$. For $\nu \in \mathbb{M}_\pi$, the Radon–Nikodym derivative of $\pi$ with respect to $\nu$ is given, for $\nu$-almost every $x$, by

$$\frac{\mathrm{d}\pi}{\mathrm{d}\nu}(x) = \frac{\pi(x)}{\nu(x)} \ .$$

For any probability measure $\nu$ and measurable function $f$, we use the shorthand

$$\nu(f) = \int_{\mathcal{X}} f(x)\nu(\mathrm{d}x) \,, \qquad \|f\|_{\infty} = \sup_{x \in \mathbb{R}^d} |f(x)| \,.$$

Given a Markov transition kernel $K$ on $(\mathbb{R}^d, \mathcal{B}(\mathbb{R}^d))$ and a probability measure $\mu$, we define

$$\mu K(\mathrm{d}y) = \int \mu(\mathrm{d}x) K(x, \mathrm{d}y) \,.$$

A Markov kernel $K$ is said to be $\pi$-invariant if $\pi K = \pi$.

The Kullback–Leibler between $\pi$ and $\mu \in \mathbb{M}_\pi$ is defined as

$$\mathsf{KL}(\pi \,\|\, \mu) = \int \log \frac{\mathrm{d}\pi}{\mathrm{d}\mu}(x) \, \pi(\mathrm{d}x) \,.$$

Finally, the total variation distance between two probability measures $\mu$ and $\nu$ is defined as

$$\|\mu - \nu\|_{\mathsf{TV}} = \sup_{A \in \mathcal{B}(\mathbb{R}^d)} |\mu(A) - \nu(A)| \,.$$

### 2.1. Contracting Sequences of Probability Measures

Our objective is to construct a sequence of probability measures $\{\mu_t\}_{t \in \mathbb{N}}$ in $\mathbb{M}_\pi$ that converges to the target distribution $\pi$ and enjoys explicit contraction guarantees. More precisely, we seek sequences satisfying, for some constant $0 \leq \rho \leq 1$ and all $t \in \mathbb{N}$,

$$\mathsf{KL}(\pi \,\|\, \mu_{t+1}) \leq \rho \, \mathsf{KL}(\pi \,\|\, \mu_t) \,. \tag{1}$$

This divergence is natural for importance sampling as it control bias and variance of importance weights, and it also is the one directly contracted by Entropic Mirror. When $\rho < 1$, this property ensures that $\mathsf{KL}(\pi \,\|\, \mu_t)$ converges geometrically fast to zero, which is crucial for rapidly reducing target-proposal mismatch and escape severe weight degeneracy regimes.

We focus on sequences generated as iterates of a mapping $\mathcal{F} : \mathbb{M}_\pi \to \mathbb{M}_\pi$,

$$\mu_{t+1} = \mathcal{F}(\mu_t) \,,$$

and refer to such a mapping as *contracting* when (1) holds. Recent works have shown that mirror descent–type mappings provide a principled way to construct such sequences (Beck & Teboulle, 2003; Dai et al., 2016; Daudel et al., 2021; Korba & Portier, 2022).

### 2.2. Entropic Mirror Descent Mapping

For all $\mu \in \mathbb{M}_\pi$ and $0 < \varepsilon \leq 1$, the Entropic Mirror Descent mapping $\mathcal{F}_{\mathsf{e}}$ is defined as

$$\mathcal{F}_{\mathsf{e}}(\mu) \propto \left(\frac{\mathrm{d}\pi}{\mathrm{d}\mu}\right)^{\varepsilon} \mu \,. \tag{2}$$

When it exists, the probability density of the associated measure is $\mathcal{F}_{\mathsf{e}}(\mu)(x) \propto \pi(x)^{\varepsilon}\mu(x)^{1-\varepsilon}$. The mapping $\mathcal{F}_{\mathsf{e}}$ corresponds to a particular instance of mirror descent applied to the minimization of the Kullback–Leibler divergence (Beck & Teboulle, 2003). It was shown in Korba & Portier (2022) that $\mathcal{F}_{\mathsf{e}}$ satisfies the contraction property (1) with $\rho = 1 - \varepsilon$. The parameter $\varepsilon$ thus controls a trade-off between the speed of convergence and the discrepancy between successive iterates.

From Jensen's inequality, it also holds that

$$\mathsf{KL}(\mu \,\|\, \mathcal{F}_{\mathsf{e}}(\mu)) \leq \varepsilon \, \mathsf{KL}(\mu \,\|\, \pi) \,,$$

which shows that consecutive iterates remain close when $\varepsilon$ is small. This observation is central in practice, as it impacts the quality of Monte Carlo approximations of the mapping.

### 2.3. Limitations of Entropic Mirror Descent

Although the Entropic Mirror Descent mapping enjoys strong theoretical guarantees, it is generally intractable for sampling and density evaluation. Practical implementations therefore rely on Monte Carlo approximations of $\mathcal{F}_{\mathsf{e}}(\mu)$, typically based on weighted samples drawn from $\mu$. As in importance sampling, the quality of these approximations heavily depends on the overlap between $\mu$ and $\pi$.

In multimodal settings, when the current iterate $\mu_t$ fails to cover certain regions of high probability mass under $\pi$, Monte Carlo approximations of $\mathcal{F}_{\mathsf{e}}(\mu_t)$ may entirely miss these regions. As a consequence, the resulting approximate sequence may fail to converge to $\pi$. This phenomenon persists even when the variational family is sufficiently rich, as illustrated in Figure 1. The target distribution is set to a univariate Gaussian mixture with two components and the initial proposal $\mu_0$ to a mixture with its two modes at 0. In this setting, the sequence $\{\mu_t\}_{t \geq 0}$ can be exactly computed and we indeed observe that it converges to $\pi$. However, when replacing the mapping $\mathcal{F}_{\mathsf{e}}$ by its Monte Carlo counterpart, the update reduces to sampling from the current proposal distribution and the method fails to visit the second mode.

These limitations motivate the introduction of additional mechanisms that explicitly promote exploration of the state space. In particular, it is natural to consider Markov transition kernels that are able to move samples into regions that are unlikely under the current proposal but relevant under the target distribution. This idea follows recent works such as Samsonov et al. (2022) which design non-local moves and global proposals to improve sampling performance of MCMC algorithms or Cabezas et al. (2024) where the authors proposed an adaptive MCMC scheme combining a non-local, flow-informed transition kernel and a local transition kernel, which generate samples from a sequence of annealed target distributions.

# 3. Entropic Mirror Monte Carlo

## 3.1. Mapping with Invariant Markov Kernels

To address the aforementioned limitations, we introduce a new mapping that combines importance reweighting with Markovian dynamics. Let $K_\pi$ be a $\pi$-invariant Markov transition kernel. For $0 < \varepsilon \le 1$ and $0 < \lambda \le 1$, we define

$$\mathcal{F}_{\mathsf{em}}(\mu;\,\lambda, K_\pi, \varepsilon) = \lambda\,\mathcal{F}_{\mathsf{e}}(\mu) + (1-\lambda)\,\mathcal{F}_{K_\pi}(\mu)\,, \quad (3)$$

where

$$\mathcal{F}_{K_\pi}(\mu) \propto \left(\frac{\mathrm{d}\pi}{\mathrm{d}\mu}\right)^\varepsilon \mu K_\pi\,.$$

The proposed mapping is a convex combination of probability measures, or of their densities when they exist. The first term preserves the contraction properties of Entropic Mirror Descent and exploits regions already identified by the current proposal. The second term leverages the exploratory power of the Markov kernel and assigns significant weight to samples that reach previously unexplored regions of $\pi$. The parameter $\lambda$ balances these effects: large values favor the mirror step but may weaken exploration, while small values favor the kernel step but may deteriorate contraction. The calibration of this parameter is therefore crucial to obtaining an efficient estimator

**Lemma 3.1.** *Given $0 < \varepsilon \le 1$ and $0 < \lambda \le 1$, if $\mu \in \mathbb{M}_\pi$ satisfies*

$$\left\|\frac{\mathrm{d}\pi}{\mathrm{d}\mu}\right\|_\infty < \infty\,,$$

*then $\mathcal{F}_{\mathsf{em}}(\mu;\,\lambda, K_\pi, \varepsilon)$ is well defined and is a probability measure in $\mathbb{M}_\pi$. Moreover, it satisfies*

$$\left\|\frac{\mathrm{d}\pi}{\mathrm{d}\mathcal{F}_{\mathsf{em}}(\mu;\,\lambda, K_\pi, \varepsilon)}\right\|_\infty < \infty\,.$$

Lemma 3.1 ensures controlled initial mismatch and well-defined reweighting. Namely, if we have an initial distribution $\mu_0$ which satisfies $\|\mathrm{d}\pi/\mathrm{d}\mu_0\|_\infty < \infty$, the mapping (3) can define a sequence of probability measures $\{\mu_t\}_{t\in\mathbb{N}}$ given, for $0 < \lambda_{t-1} \le 1$; $0 < \varepsilon \le 1$, by

$$\mu_t = \mathcal{F}_{\mathsf{em}}(\mu_{t-1};\,\lambda_{t-1}, K_\pi, \varepsilon)\,, \quad (4)$$

since all the iterates $\mu_t$ then also satisfy $\|\mathrm{d}\pi/\mathrm{d}\mu_t\|_\infty < \infty$. Note that such a condition does not prevent finite-sample weight degeneracy.

**Proposition 3.2.** *Let $0 < \varepsilon \le 1$ and $\mu_0 \in \mathbb{M}_\pi$ such that $\|\mathrm{d}\pi/\mathrm{d}\mu_0\|_\infty < \infty$. If $K_\pi$ is $\pi$-invariant, then there exists a sequence $0 < \lambda_t \le 1$ such that the iterates*

$$\mu_t = \mathcal{F}_{\mathsf{em}}(\mu_{t-1};\,\lambda_{t-1}, K_\pi, \varepsilon)$$

*satisfy, for all $t \in \mathbb{N}$,*

$$\mathsf{KL}(\pi \parallel \mu_t) \le (1-\varepsilon)^t \mathsf{KL}(\pi \parallel \mu_0)\,.$$

Proposition 3.2 establishes that the contraction property is preserved. This result also establishes that if we have not achieved $\mu_t = \pi$ and set $\epsilon = 1$, we can always find a sequence $\{\lambda_t\}_{t\in\mathbb{N}}$ such that, for any choice of $\pi$-invariant transition kernel $K_\pi$, the mapping $\mathcal{F}_{\mathsf{em}}$ differs from the original Entropic Mirror Descent transformation (2). For simplicity we state the results for fixed $\varepsilon$, although the arguments extend directly to time-varying choices $\varepsilon_t \in (0, 1]$.

## 3.2. Mapping with Unadjusted Langevin Kernel

Proposition 3.2 only requires that $K_\pi$ is $\pi$-invariant, so our mapping can be used with various Markov Chain Monte Carlo methods such as Metropolis-adjusted Langevin (MALA, Besag, 1994), Hamiltonian Monte Carlo (HMC, Neal et al., 2011) or iSIR (Andrieu et al., 2010) algorithms.

In practice, however, Markov kernels that are not $\pi$-invariant may exhibit strong exploratory properties. A prominent example is the Unadjusted Langevin Algorithm, (ULA, Roberts & Tweedie, 1996). This Markov Chain Monte Carlo algorithm is deduced from the Euler-Maruyama discretization of the Langevin diffusion which yields the iterates:

$$X_{k+1} = X_k + \gamma\nabla\log\pi(X_k) + \sqrt{2\gamma}Z_{k+1}\,,$$

where $(Z_k)_{k\ge1}$ is an *i.i.d.* sequence of $d$-dimensional standard Gaussian vectors and $\gamma$ is a fixed positive step-size. We denote by $\mathsf{R}_\gamma$ the Markov kernel associated with a single step of the Unadjusted Langevin Algorithm with step size $\gamma > 0$. For all $x \in \mathbb{R}^d$, $\mathsf{R}_\gamma(x, \cdot)$ is the Gaussian distribution with mean $x + \gamma\nabla\log\pi(x)$ and covariance matrix $2\gamma\mathbf{I}_d$. Although $\mathsf{R}_\gamma$ does not admit $\pi$ as invariant distribution, it is known to possess a unique invariant distribution, denoted by $\pi_\gamma$, under suitable regularity assumptions.

*Condition* 3.1. The target distribution $\pi$ satisfies a log-Sobolev inequality: there exists a constant $C_{\mathrm{LS}} > 0$ such that for all smooth functions $g : \mathbb{R}^d \to \mathbb{R}$,

$$\int g^2 \log\frac{g^2}{\int g^2\mathrm{d}\pi}\,\mathrm{d}\pi \le C_{\mathrm{LS}}\int\|\nabla g\|^2\,\mathrm{d}\pi\,.$$

*Condition* 3.2. The log-density of the target distribution is continuously differentiable and $L$-smooth: there exists a constant $L > 0$ such that for all $x, y \in \mathbb{R}^d$,

$$\|\nabla\log\pi(x) - \nabla\log\pi(y)\| \le L\|x - y\|\,.$$

Conditions 3.1 and 3.2 are standard in the sampling literature (Durmus & Moulines, 2019; Vempala & Wibisono, 2019; Chewi et al., 2022; Erdogdu et al., 2022). In particular, Condition 3.1 provides a powerful tool for analyzing the exponential convergence of Markov semi-groups (Bakry et al., 2014). This condition holds for a large class of probability measures, including log-concave distributions and is stable under bounded perturbations (Holley & Stroock, 1987).

*Condition* 3.3. There exist a measurable function $V : \mathbb{R}^d \to [1, +\infty)$, a constant $b \geq 0$, and a compact set $C \subset \mathbb{R}^d$ such that

$$\mathsf{R}_\gamma V \leq V - 1 + b \mathbb{1}_C . \tag{5}$$

Condition 3.3 ensures the existence of an invariant distribution $\pi_\gamma$ for $\mathsf{R}_\gamma$ (Meyn & Tweedie, 2012). Using only Condition 3.2, it is easily shown that for any compact set $\mathsf{K} \subset \mathbb{R}^d$ there exists $C \geq 0$ such that for all $x \in \mathsf{K}$ and $A \in \mathcal{B}(\mathbb{R}^d)$

$$\mathsf{R}_\gamma(x, A) \geq C_\mathsf{K} \mathrm{Leb}(A \cap \mathsf{K}) , \tag{6}$$

and thus $\mathsf{R}_\gamma$ is Leb-irreducible and strongly aperiodic. Therefore, following Theorem 14.0.1 of Meyn & Tweedie (2012), $\mathsf{R}_\gamma$ is positive recurrent and admits a unique invariant distribution if and only if (5) is satisfied for some petite set $C$. In Condition 3.3, we only strengthen this condition on $C$ and suppose that it is compact which is satisfied in most applications.

We consider the sequence of probability measures $\{\mu_t^\mathsf{R}\}_{t \in \mathbb{N}} \subset \mathbb{M}_\pi$ defined recursively, for $t \geq 1$, by

$$\mu_t^\mathsf{R} = \mathcal{F}_\mathsf{em}\left(\mu_{t-1}^\mathsf{R}; \lambda_{t-1}, \mathsf{R}_\gamma, \varepsilon\right) ,$$

with initial distribution $\mu_0^\mathsf{R} \in \mathbb{M}_\pi$.

**Theorem 3.3.** *Assume Conditions 3.1, 3.2, and 3.3 hold. Let $0 < \gamma \leq \left(2C_\mathrm{LS}L^2\right)^{-1}$. Let $\mu_0^\mathsf{R} \in \mathbb{M}_\pi$ satisfy $\left\|\mathrm{d}\pi/\mathrm{d}\mu_0^\mathsf{R}\right\|_\infty < \infty$. Then, there exists a sequence $0 < \lambda_t \leq 1$ such that*

$$\left\|\pi - \mu_t^\mathsf{R}\right\|_\mathsf{TV} \leq (1 - \varepsilon)^{t/2}\sqrt{2\,\mathsf{KL}\left(\pi_\gamma \parallel \mu_0^\mathsf{R}\right)}$$
$$+ 4\sqrt{2\gamma dL^2 C_\mathrm{LS}} .$$

Theorem 3.3 separates contraction toward the ULA invariant distribution $\pi_\gamma$ induced by the Entropic Mirror component (first right term), from the discretization bias to $\pi$, controlled by Proposition A.1(second right term).

### 3.3. Monte Carlo Implementation: the EM2C Algorithm

In most settings the ideal map $\mathcal{F}_\mathsf{em}$ is intractable and can only be accessed through samples. Algorithm 1 replaces the oracle sequence (4) by surrogate proposals $(\tilde{\mu}_t)_{t \in \mathbb{N}}$ obtained via Monte Carlo sampling, weighting, resampling, and projection onto a tractable parametric family. Namely, given the current proposal, say $\tilde{\mu}_t$, we sample $(X_t^{1:N})$ from $\tilde{\mu}_t$ and propagate them through a Markov kernel $(Y_t^{1:N})$, which yields the empirical distribution

$$\widehat{\mathcal{F}}_{\tilde{\mu}_t}^N = \lambda_t \sum_{i=1}^N \omega_t^i \delta_{X_t^i} + (1 - \lambda_t) \sum_{i=1}^N \varpi_t^i \delta_{Y_t^i} .$$

---

**Algorithm 1** Entropic Mirror Monte Carlo (EM2C)

**Input:** number of iterations $T$, number of samples $N$, initial distribution $\mu_0$, parameters $0 < \varepsilon \leq 1, 0 < \lambda_t \leq 1$ ($0 \leq t \leq T - 1$), Markov kernel $K$.
**Output:** proposal distribution $\tilde{\mu}_T$.
**for** $t = 0$ **to** $T - 1$ **do**
  Sample $X_t^{1:N} \sim \tilde{\mu}_t^{\otimes N}$.
  Sample $Y_t^{1:N} \sim K(X_t^1, \cdot) \otimes \cdots \otimes K(X_t^N, \cdot)$.
  **for** $i = 1$ **to** $N$ **do**
    Compute

$$\omega_t^i \propto \left\{\frac{\mathrm{d}\pi}{\mathrm{d}\tilde{\mu}_t}(X_t^i)\right\}^\varepsilon , \quad \varpi_t^i \propto \left\{\frac{\mathrm{d}\pi}{\mathrm{d}\tilde{\mu}_t}(Y_t^i)\right\}^\varepsilon .$$

  **end for**
  **for** $i = 1$ **to** $N$ **do**
    Sample

$$Z_t^i \sim \widehat{\mathcal{F}}_{\tilde{\mu}_t}^N = \lambda_t \sum_{j=1}^N \omega_t^j \delta_{X_t^j} + (1 - \lambda_t) \sum_{j=1}^N \varpi_t^j \delta_{Y_t^j}.$$

  **end for**
  Compute

$$\tilde{\mu}_{t+1} = \underset{\mu_\theta \in \mathsf{F}_\theta}{\mathrm{argmin}} \, \mathcal{R}\left(\widehat{\mathcal{F}}_{\tilde{\mu}_t}^N, \mu_\theta\right).$$

**end for**

---

The empirical measure $\widehat{\mathcal{F}}_{\tilde{\mu}_t}^N$ cannot be directly reused as the next proposal since propagated particles typically lie outside the support of a purely empirical measure, making subsequent density ratios ill-defined. We therefore project onto a tractable parametric family $\mathsf{F}_\theta$ to allow importance weights computation. The surrogate distribution within Algorithm 1 is then computed as the minimizer of a loss function $\mathcal{R}$ between $\mu_\theta \in \mathsf{F}_\theta$ and the empirical measure $\widehat{\mathcal{F}}_{\tilde{\mu}_t}^N$.

The projection onto $\mathsf{F}_\theta$ can be achieved in several ways. We begin by describing two common situations, both of which consist to choosing $\mathcal{R}$ as the Kullback–Leibler, and minimizing, with respect to $\theta$, the Kullback–Leibler divergence between the sample distribution and the distribution $\mu_\theta$.

**Mixture models and EM algorithm.** For mixture models with base probability distributions $\{f_{\eta_k} ; 1 \leq k \leq K\}$, where $\eta_k$ denotes the $k$-th component parameter, we have:

$$\mathsf{F}_\theta = \left\{\sum_{k=1}^K p_k f_{\eta_k} ; \sum_{k=1}^K p_k = 1 , p_k \geq 0\right\} .$$

Estimation is typically carried out via the Expectation Maximization (EM) algorithm, which produces a sequence of

parameter estimates by maximizing approximately the log-likelihood function:

$$\mathcal{R}\left(\widehat{\mathcal{F}}^N_{\tilde{\mu}_t}, \mu_\theta\right) = \sum_{i=1}^N \log\left(\sum_{k=1}^K p_k \, f_{\eta_k}(Z^i_t)\right) \ ,$$

where $Z^{1:N}_t$ are samples from $\widehat{\mathcal{F}}^N_{\tilde{\mu}_t}$.

**Normalizing flows.** In recent years, several works have introduced hybrid methods which combine normalizing flows with MCMC algorithms to enhance sampling performance (Gabrié et al., 2022; Grenioux et al., 2023). Normalizing flows aim at learning a flexible parametric distribution from $Z^{1:N}_t \sim \widehat{\mathcal{F}}^N_{\tilde{\mu}_t}$ by constructing an invertible and differentiable transformation that maps a simple base distribution to the distribution of $Z^{1:N}_t$. Consider a family of bijective mappings $f_\theta : \mathbb{R}^d \to \mathbb{R}^d$ parameterized by $\theta \in \Theta$, such that a random variable $Z$ drawn from a tractable base density $p_Z$ is transformed into $X = f_\theta(Z)$ with induced density given by the change-of-variables formula

$$p_\theta(x) = p_Z\big(f^{-1}_\theta(x)\big) \left|\det D f^{-1}_\theta(x)\right| \ ,$$

where $D f^{-1}_\theta$ is the Jacobian of $f^{-1}_\theta$. Therefore,

$$\mathsf{F}_\theta = \left\{x \mapsto p_Z\big(f^{-1}_\theta(x)\big) \left|\det D f^{-1}_\theta(x)\right| \ ; \ \theta \in \Theta\right\} \ .$$

and the log-likelihood objective is then given by

$$\mathcal{R}\left(\widehat{\mathcal{F}}^N_{\tilde{\mu}_t}, \mu_\theta\right) = \sum_{i=1}^N \log p_\theta(Z^i_t) \ ,$$

which can be equivalently expressed as

$$\mathcal{R}\left(\widehat{\mathcal{F}}^N_{\tilde{\mu}_t}, \mu_\theta\right) = \sum_{i=1}^n \log p_Z\big(f^{-1}_\theta(Z^i_t)\big)$$
$$+ \sum_{i=1}^n \log\left|\det D f^{-1}_\theta(Z^i_t)\right| \ .$$

Although the projection is essential for implementation, the contraction guarantees for the ideal $\mathcal{F}_{em}$ do not generally extend to the projected sequence $(\tilde{\mu}_t)_{t\in\mathbb{N}}$ used in practice. The approximation error, that depends on the expressiveness of $\mathsf{F}_\theta$ and the numerical procedure employed, may weaken or break the contraction. In particular, if $\mathsf{F}_\theta$ lacks the relevant multimodal structure, the projection can collapse distinct modes, even when weighted particles successfully reach them.

Nevertheless, our experiments show that Algorithm 1 remains stable and effective in practice.

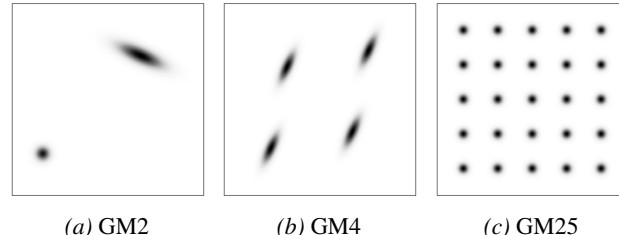

| *(a)* GM2 | *(b)* GM4 | *(c)* GM25 |

*Figure 2.* Marginal Gaussian mixture distribution $\widetilde{\pi}_i$ associated with the benchmark target distributions $\pi_{i,d}$ as defined in (7).

## 4. Numerical Study

In this section, we present numerical experiments[1] assessing the performance of Algorithm 1 on multimodal target distributions. We evaluate mode recovery, the effect of $\lambda$ and the exploration kernel, and the discrepancy between the learned proposal $\tilde{\mu}_T$ and the target distribution $\pi$ using : sliced Wasserstein distance, energy distance, and negative log-likelihood. Definitions and computation are provided in Appendix E.1–E.3. All methods are run under comparable computational budgets measured in terms of target density and gradient evaluations (Appendix E.5). For EM2C, the number of kernel evaluations is slightly reduced to account for the additional cost of the projection step.

### 4.1. Multidimensional Gaussian Mixture Benchmarks

**Target distributions.** We first consider targets built as a tensor product of one of the following two-dimensional Gaussian mixture models (see Figure 2).

- **GM2 (two-component, unbalanced)**

$$\widetilde{\pi}_1 = 0.2\mathcal{N}(\mathbf{0}_2, I_2) + 0.8\mathcal{N}\left(\begin{bmatrix}20\\20\end{bmatrix}, \begin{bmatrix}10 & -4\\-4 & 3\end{bmatrix}\right) \ .$$

- **GM4 (four-component, anisotropic)**

$$\widetilde{\pi}_2 = 0.25\sum_{j=1}^4 \mathcal{N}\left(m_j, \begin{bmatrix}3 & 4\\4 & 10\end{bmatrix}\right) \ ,$$

with $m_1 = (-10, 10)^\top$, $m_2 = (10, -10)^\top$, $m_3 = (15, 15)^\top$, and $m_4 = (-15, -15)^\top$.

- **GM25 (twenty-five-component, isotropic)**

$$\widetilde{\pi}_3 = \frac{1}{25}\sum_{\ell=0}^4 \sum_{k=0}^4 \mathcal{N}\left(\begin{bmatrix}5\ell\\5k\end{bmatrix}, 0.25 I_2\right) \ .$$

Given an even dimension $d$ and a mixture distribution $\widetilde{\pi}_i$, the target writes as

$$\pi_{i,d} = \widetilde{\pi}_i^{\otimes d/2}, \quad i = 1, 2, 3 \ . \tag{7}$$

[1]The code used is available at https://github.com/jstoehr/EM2C.

**Experimental design.** In all experiments, the initial proposal is a Gaussian distribution $\mu_0 = \mathcal{N}(30\,\mathbf{1}_d, I_d)$, placing the initial mass in a low-probability region of the target and providing a common, challenging initialization across all models.

The variational family $\mathsf{F}_\theta$ is chosen as a tensorized Gaussian mixture model, and the projection step is implemented via an expectation–maximization (EM) algorithm. Implementation details, including the structure of the variational family and EM hyperparameters, are provided in Appendix B.4.

Algorithm 1 is run with $\varepsilon = 0.8$, $N = 2{,}000$ particles per iteration, and constant mixing parameters $\lambda \in \{0.5, 0.8, 1.0\}$. The number of EM2C iterations $T$ and the parameters of the exploration kernel $K$ are tuned to the target model and the ambient dimensions $d$ to account for differences in scale and geometry of the target $\pi$.

For each setting, the exploration kernel $K$ is chosen either as a $\pi$-invariant Random Walk (RW) Metropolis kernel, with variance parameter $\sigma_K^2$, or as ULA, with step size $\gamma_K$. At each EM2C iteration, the selected kernel is applied for a fixed number of $n_K$ steps. Complete kernel configurations and experimental settings for all targets and dimensions are given in Appendix B.4.

**Results.** Figure 3 illustrates the evolution of the proposal distribution obtained with Algorithm 1 with respect to the GM4 target in dimension $d = 10$. For each value of $\lambda$, we compare the behaviour of EM2C using RW or ULA kernel. It shows the impact of both the mixing parameter and the exploration kernel on the preservation of multimodality: smaller values of $\lambda$ promote exploration and help maintaining multiple modes, while larger values may result in mode imbalance when exploration is insufficient. Additional qualitative evolution plots for the other mixture targets and dimensions are provided in Appendix B.2. Overall, the ULA kernel consistently outperforms the RW kernel, as the latter may suffer from frequent rejections in low-density regions, whereas ULA leverages gradient information to guide proposals toward regions of high target density.

This qualitative behaviour is further reflected in the evolution of the sliced Wasserstein distance across EM2C iterations. For the GM4 benchmark with $d = 10$, Figure 4 shows that smaller values of $\lambda$ lead to a rapid reduction of the discrepancy between the proposal and the target, whereas $\lambda = 1$ results in poor convergence. Final sliced Wasserstein distances at convergence for the GM4 benchmark with dimensions $d = 10$ and $d = 20$ are reported in Table 1. Additional sliced Wasserstein results for the other targets and dimensions are provided in Appendix B.1.

Appendix B.3 gives a matched-budget comparison of EM2C with standard MCMC on the GM4 benchmark ($d = 10$).

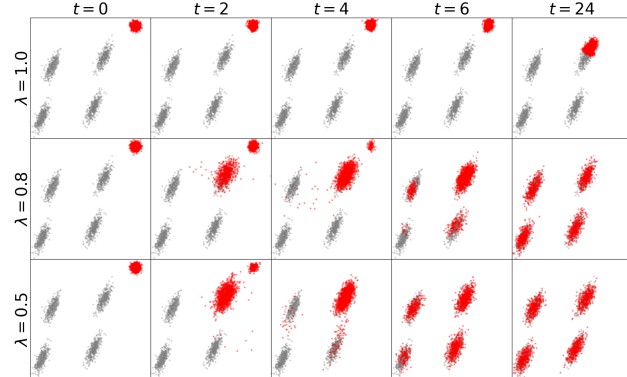

*(a)* Unadjusted Langevin exploration kernel

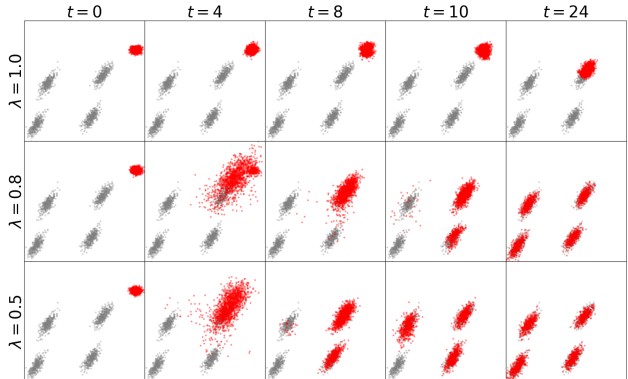

*(b)* Random Walk Metropolis–Hastings exploration kernel

*Figure 3.* Evolution of EM2C proposal distributions for the GM4 target ($d = 10$). Rows correspond to $\lambda \in \{0.5, 0.8, 1.0\}$, columns to EM2C iterations. Gray and red points are samples from $\pi$ and $\tilde{\mu}_t$, respectively. The marginal samples on the first two coordinates are displayed.

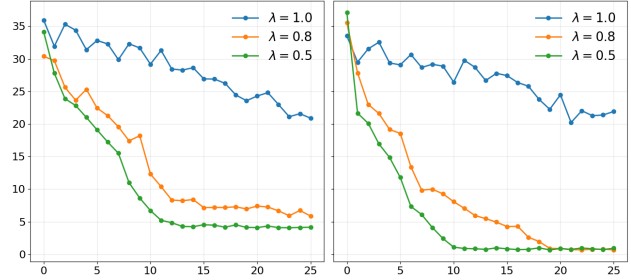

*Figure 4.* Evolution of the sliced Wasserstein distance over EM2C iterations for the GM4 target ($d = 10$), shown for Random Walk (left) or Unadjusted Langevin (right) exploration kernels and $\lambda \in \{0.5, 0.8, 1.0\}$.

**High-dimensional regime with fixed number of modes.** The tensorized targets in (7) have a number of modes growing exponentially with the dimension, which limits scalability studies in very high dimension. To isolate the effect of ambient dimensionality, we consider an additional Gaussian mixture benchmark in dimension $d = 200$ with

*Table 1.* Final sliced Wasserstein distances of EM2C for the GM4 benchmark with dimensions $d = 10$ and $20$, using Random Walk (RW) or Unadjusted Langevin Algorithm (ULA) exploration kernels. We report the mean $\pm$ standard deviation over three independent runs.

| $d$ | $\lambda$ | RW | ULA |
|---|---|---|---|
| 10 | 1.0 | $20.60 \pm 0.40$ | $20.41 \pm 1.07$ |
| 10 | 0.8 | $4.42 \pm 2.64$ | $0.81 \pm 0.17$ |
| 10 | 0.5 | $2.09 \pm 1.80$ | $0.84 \pm 0.03$ |
| 20 | 1.0 | $24.50 \pm 1.08$ | $23.28 \pm 0.70$ |
| 20 | 0.8 | $7.82 \pm 0.52$ | $5.37 \pm 0.68$ |
| 20 | 0.5 | $1.81 \pm 0.58$ | $3.13 \pm 0.72$ |

only $K = 4$ modes, and where multimodality lies in a low-dimensional subspace and the remaining coordinates correspond to isotropic Gaussian noise (see Appendix B.5).

Results in Appendix B.5 show that EM2C recovers the modes despite the challenging initialization and high ambient dimension, highlighting its robustness to irrelevant dimensions.

### 4.2. Multimodal Targets with Complex Geometry

We further assess the performance of EM2C on a collection of target distributions exhibiting strong multimodality and nontrivial geometry. Such benchmarks are commonly used to stress-test sampling algorithms, as they combine disconnected modes, curved supports, and regions of low probability that hinder exploration.

**Target distributions.** We study the following target distributions on $\mathbb{R}^2$.

- **Dual Moons**: two interleaving half-circles with additive Gaussian noise. For $x = (x_1, x_2)$, $(a, b) = (0.09, 0.08)$

$$\pi(x) = \left(1 + e^{4x_1/a}\right) \exp\left\{-\frac{(\|x\| - 1)^2}{b} - \frac{(x_1 - 2)^2}{2a}\right\} .$$

- **Two Rings**: two concentric annular distributions with different radii. For $\sigma = 0.1$,

$$\pi(x) = \frac{1}{\|x\|} \sum_{k \in \{1, 4\}} \exp\left(-\frac{(\|x\| - k)^2}{2\sigma^2}\right) .$$

**Experimental design.** We evaluate EM2C under challenging initializations, where the initial distribution places little on the target support or covers only a subset of the modes. Such settings arise when prior information is limited or misspecified and make exploration essential. By contrast, initializing from a broad distribution already covering the target may allow standard samplers to perform well,

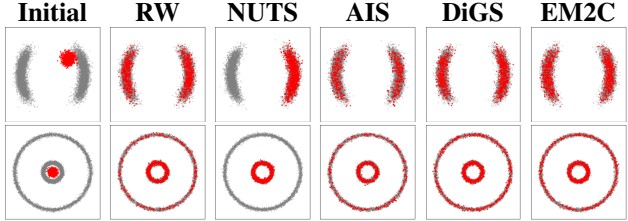

*Figure 5.* Qualitative comparison of sampling methods on two-dimensional multimodal targets: Dual Moons (top row), Two Rings (bottom row). Gray and red points are samples from $\pi$ and $\tilde{\mu}_t$, respectively.

but mask the intrinsic difficulty of multimodal sampling. We therefore focus on regimes where exploration is critical and where combining entropic reweighting with Markovian exploration provides a tangible advantage.

In Algorithm 1, the projection step is achieved using normalizing flows as the variational family $\mathsf{F}_\theta$ (see Appendix C.2.1 for details).

We compare EM2C against (i) Random Walk MCMC Sampler (RW), (ii) the No-U-Turn Sampler (NUTS, Hoffman & Gelman, 2014), (iii) Annealed Importance Sampling (AIS, Neal, 2001), and (iv) Diffusive Gibbs Sampling (DIGS, Chen et al., 2024), a diffusion-based MCMC method combining Gaussian perturbation steps with local Langevin dynamics to improve transitions between distant modes. For a given target, all methods share the same initialization and number of particles ($N = 10{,}000$). Complete implementation details and hyperparameters are provided in Appendix C.2.

**Results.** Figure 5 presents qualitative density estimates obtained by each method at convergence. Under the challenging initialization considered here, the behaviour of the compared methods differs markedly. Random-Walk MCMC and AIS both recover the main modes of the target distributions under the considered initialization, with AIS generally providing lower discrepancies than RW. In contrast, NUTS typically concentrates on a single mode in this setting and fails to explore the full target support. DiGS also performs strongly on these benchmarks, achieving low discrepancies and accurate recovery of the target geometry. In comparison, EM2C yields the lowest discrepancies across both benchmarks while maintaining robust mode coverage under the considered initialization. Additional qualitative results illustrating the evolution of EM2C proposals on two-dimensional benchmarks are provided in Appendix C.

These qualitative observations are corroborated by quantitative evaluations based on both the sliced Wasserstein distance and the energy distance. Table 2 reports the final $\mathrm{SW}_2(\mu_T, \pi)$ and $\mathrm{ED}(\mu_T, \pi)$ values for all considered datasets and methods. EM2C achieves systematically lower discrepancies under challenging initialization regimes, con-

*Table 2.* Sliced Wasserstein distance (SW$_2$) and energy distance (ED) on two-dimensional benchmarks. We report the mean $\pm$ standard deviation over three runs.

*(a)* Dual Moons

|  | RW | NUTS | AIS | DiGS | EM2C |
|---|---|---|---|---|---|
| SW$_2$ | $0.62 \pm 0.01$ | $1.66 \pm 0.05$ | $0.34 \pm 0.19$ | $0.074 \pm 0.037$ | $\mathbf{0.071 \pm 0.016}$ |
| ED | $0.082 \pm 0.011$ | $1.40 \pm 0.01$ | $0.06 \pm 0.07$ | $0.0005 \pm 0.0003$ | $\mathbf{0.0004 \pm 0.0001}$ |

*(b)* Two Rings

|  | RW | NUTS | AIS | DiGS | EM2C |
|---|---|---|---|---|---|
| SW$_2$ | $0.55 \pm 0.01$ | $1.41 \pm 0.001$ | $0.22 \pm 0.18$ | $0.053 \pm 0.011$ | $\mathbf{0.049 \pm 0.008}$ |
| ED | $0.056 \pm 0.002$ | $0.434 \pm 0.001$ | $0.009 \pm 0.067$ | $0.0008 \pm 0.0002$ | $\mathbf{0.0006 \pm 0.0002}$ |

firming its improved ability to explore complex target geometries.

When initialized from a sufficiently broad proposal covering the target support, the performance gap between methods narrows, as exploration becomes less critical. In the more challenging and practically relevant settings considered here, EM2C nevertheless exhibits superior robustness to initialization and improved mode recovery.

### 4.3. Bayesian Neural Network Posterior

We evaluate EM2C on Bayesian neural network (BNN) posterior sampling, a high-dimensional multimodal and complex inference task, following the setup of Chen et al. (2024).

**Target distribution.** Let $\mathcal{D}_{\text{train}} = \{(x_i, y_i)\}_{i=1}^N$ be a dataset. The posterior over network parameters $\theta \in \mathbb{R}^d$ is defined as

$$\pi(\theta) = p(\theta \mid \mathcal{D}_{\text{train}}) \propto p(\theta) \prod_{i=1}^N p(y_i \mid x_i, \theta) .$$

We use a one-hidden-layer ReLU network

$$f_\theta(x) = W_2^\top \sigma\left(d_x^{-1/2} x^\top W_1 + 0.1 b_1\right),$$

with $\theta = (W_1, b_1, W_2)$, $d_x = 20$, $d_h = 25$, and parameter dimension $d = 550$. The prior is $\mathcal{N}(0, I)$ and the likelihood $p(y \mid x, \theta)$ is Gaussian with noise $\sigma_n = 0.1$.

**Experimental design.** We generate data as in Chen et al. (2024) using synthetic datasets generated from a ground-truth parameter $\theta^\star \sim p(\theta)$, and compare EM2C with MALA and DiGS. MALA also serves as the underlying Langevin transition used in both EM2C and DiGS. EM2C uses a Gaussian mixture with 20 components and diagonal-covariances for projection. All methods are initialized from the prior and use $N = 500$ samples for evaluation. Detailed implementation and hyperparameters are provided in Appendix D.

*Table 3.* Test negative log-likelihood (NLL) for Bayesian neural network posterior sampling. Mean $\pm$ standard deviation over 10 runs.

| Method | MALA | DiGS | EM2C |
|---|---|---|---|
| NLL | $0.2219 \pm 0.0170$ | $0.2016 \pm 0.0128$ | $\mathbf{0.1294 \pm 0.0707}$ |

**Results.** We evaluate performance using the negative log-likelihood (NLL) on a held-out test set, computed as the average test log-likelihood under parameters sampled from the learned distribution $\tilde{\mu}_T$ (see Appendix E.3 for details). Results are summarized in Table 3.

EM2C achieves the lowest negative log-likelihood, indicating improved posterior exploration and predictive performance. While DiGS already provides a strong baseline on this task, the additional adaptive reweighting mechanism in EM2C further enhances mode exploration.

The larger standard deviations of EM2C in Table 3 reflect its population-based proposal updates: they improve exploration but introduce additional fluctuations compared with MALA and DiGS, which average samples along a small number of long Markov chains.

Overall, EM2C provides a favorable trade-off between exploration and stability, achieving improved predictive performance while remaining robust across runs.

## 5. Discussion

In this paper, we propose a new algorithm to build sequentially proposal distributions combining a contractive Entropic Mirror Descent step with an exploratory Markov kernel component. When using the Unadjusted Langevin kernel, we show that the bias induced by the discretization remains controlled and does not hinder convergence, while preserving the contraction properties of the method. We also introduce an importance sampling scheme to approximate the idealized algorithm and demonstrate its effectiveness numerically. The main limitations and directions for future work include the deployment of the method in challenging very high-dimensional settings, notably with deep learning architectures, and the theoretical analysis of contraction properties for the Monte Carlo procedure. Another important direction concerns the practical tuning of $\lambda_t$. Indeed, the existence results are constructive only at the proof level, since admissible values depend on intractable expectations. Our result is nevertheless important as it establishes the existence of a range of $\lambda_t$ ensuring contraction. The optimal tuning of $\lambda_t$, jointly with other hyperparameters, remains challenging and is left for future work.

## Acknowledgements

Via Julien Stoehr, this project has received funding from the European Union (ERC-2022-SYGOCEAN-101071601). Views and opinions expressed are however those of the authors only and do not necessarily reflect those of the European Union or the European Research Council Executive Agency. Neither the European Union nor the granting authority can be held responsible for them.

## Impact Statement

This paper presents work whose goal is to advance the field of Machine Learning. There are many potential societal consequences of our work, none which we feel must be specifically highlighted here.

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

# A. Proofs

## A.1. Proof of Lemma 3.1

If $\|\mathrm{d}\pi/\mathrm{d}\mu\|_\infty < \infty$ then

$$\int \left(\frac{\mathrm{d}\pi}{\mathrm{d}\mu}\right)^\varepsilon \mathrm{d}\mu K_\pi < \infty;$$

so $\mathcal{F}_{\mathsf{em}}(\mu;\,\lambda, K_\pi, \varepsilon)$ is indeed a probability measure. Next, since $0 \le \varepsilon \le 1$ and $0 < \lambda \le 1$, we have that

$$\sup_{x\in\mathbb{R}^d} \frac{\mathrm{d}\pi}{\mathrm{d}\mathcal{F}_{\mathsf{em}}(\mu;\,\lambda, K_\pi, \varepsilon)}(x) \le \sup_{x\in\mathbb{R}^d} \left\{ \frac{1}{\lambda} \frac{\mathrm{d}\pi}{\mathrm{d}\mu}(x)^{1-\varepsilon} \int \frac{\mathrm{d}\pi}{\mathrm{d}\mu}(y)^\varepsilon \mu(\mathrm{d}y) \right\} < \infty \; .$$

## A.2. Proof of Proposition 3.2

For any sequence $0 < \lambda_t \le 1$, $t \in \mathbb{N}$, the convexity of the Kullback-Leibler divergence leads to

$$\begin{aligned}
\mathsf{KL}(\pi \parallel \mu_{t+1}) &\le \lambda_t \mathsf{KL}(\pi \parallel \mathcal{F}_{\mathsf{e}}(\mu_t)) + (1-\lambda_t)\mathsf{KL}(\pi \parallel \mathcal{F}_{K_\pi}(\mu_t)) \\
&\le \lambda_t(1-\varepsilon)\mathsf{KL}(\pi \parallel \mu_t) + (1-\lambda_t)\left\{ \mathsf{KL}(\pi \parallel \mu_t K_\pi) - \varepsilon\mathsf{KL}(\pi \parallel \mu_t) \right\} \\
&\quad + \lambda_t \log \int f_t^\varepsilon \mathrm{d}\mu_t + (1-\lambda_t) \log \int f_t^\varepsilon \mathrm{d}\mu_t K_\pi \; ,
\end{aligned}$$

where $f_t$ stands for the Radon-Nikodym derivative of $\pi$ with respect to $\mu_t$. The Data Processing inequality of (van Erven & Harremos, 2014) combined with the $\pi$ invariance for $K_\pi$ yield

$$\mathsf{KL}(\pi \parallel \mu_t K_\pi) = \mathsf{KL}(\pi K_\pi \parallel \mu_t K_\pi) \le \mathsf{KL}(\pi \parallel \mu_t) \; ,$$

and so

$$\mathsf{KL}(\pi \parallel \mu_{t+1}) \le (1-\varepsilon)\mathsf{KL}(\pi \parallel \mu_t) + \lambda_t \log \int f_t^\varepsilon \mathrm{d}\mu_t + (1-\lambda_t) \log \int f_t^\varepsilon \mathrm{d}\mu_t K_\pi \; .$$

Hence, any sequence $\{\lambda_t\}_{t\in\mathbb{N}}$ such that, for all $t \in \mathbb{N}$

$$\lambda_t \log \int f_t^\varepsilon \mathrm{d}\mu_t + (1-\lambda_t) \log \int f_t^\varepsilon \mathrm{d}\mu_t K_\pi \le 0 \tag{8}$$

ensures the contraction. Let $0 < \beta \le 1$ and consider the sequence

$$\lambda_t = \begin{cases} \dfrac{\log \int f_t^\varepsilon \mathrm{d}\mu_t K_\pi}{\log \int f_t^\varepsilon \mathrm{d}\mu_t K_\pi - \log \int f_t^\varepsilon \mathrm{d}\mu_t} & \text{if} \quad \log \int f_t^\varepsilon \mathrm{d}\mu_t K_\pi > 0, \\ \beta & \text{otherwise.} \end{cases} \tag{9}$$

This sequence takes values between 0 and 1. Indeed, Jensen's inequality yields that $\int f_t^\varepsilon \mathrm{d}\mu_t \le 1$, with equality solely if $\mu_t = \pi$ or $\varepsilon = 1$. Therefore, $\lambda_t$ is also less or equal to 1 when $\log \int f_t^\varepsilon \mathrm{d}\mu_t K_\pi > 0$. The results thus follows since the sequence (9) satisfies condition (8).

## A.3. Intermediate Technical Result

Before proving Theorem 3.3, we need an intermediate result.

**Proposition A.1.** *Assume Conditions 3.1, 3.2, and 3.3 and $0 < \gamma \le \left(2C_{\mathrm{LS}}L^2\right)^{-1}$. Then $\mathsf{R}_\gamma$ has a unique stationary distribution $\pi_\gamma$ that satisfies*
$$\mathsf{KL}(\pi_\gamma \parallel \pi) \le 4\gamma dL^2 C_{\mathrm{LS}} \; .$$

*Proof.* As aforementioned, (6) implies that any compact set is small for $\mathsf{R}_\gamma$ and the Lebesgue measure is an irreducibility measure. Together with Condition 3.3 and Theorem 14.0.1 from (Meyn & Tweedie, 2012), this implies that for $\pi_\gamma$-almost-everywhere $x \in \mathbb{R}^d$,
$$\lim_{\ell\to\infty} \left\| \mathsf{R}_\gamma^\ell(x, \cdot) - \pi_\gamma \right\|_{\mathsf{TV}} = 0 \; .$$

Furthermore, for all $x \in \mathbb{R}^d$ and $A \in \mathcal{B}(\mathbb{R}^d)$ such that $\mathrm{Leb}(A) > 0$, we have $\mathsf{R}_\gamma(x, A) > 0$; so $\pi_\gamma(A) = \pi_\gamma \mathsf{R}_\gamma(A) > 0$ and so $\pi_\gamma \gg \mathrm{Leb}$. Finally, for any $\nu \in \mathbb{M}_1$

$$\left\| \nu \mathsf{R}_\gamma^\ell - \pi_\gamma \right\|_{\mathsf{TV}} \leq \int \nu(\mathrm{d}x) \left\| \mathsf{R}_\gamma^\ell(x, \cdot) - \pi_\gamma \right\|_{\mathsf{TV}} ,$$

and by the Lebesgue dominated convergence theorem it follows that

$$\lim_{\ell \to \infty} \left\| \nu \mathsf{R}_\gamma^\ell - \pi_\gamma \right\|_{\mathsf{TV}} = 0 . \tag{10}$$

In (Vempala & Wibisono, 2019) (Theorem 1), it is shown that for all $\ell > 0$

$$\mathsf{KL}\!\left( \nu \mathsf{R}_\gamma^\ell \,\|\, \pi \right) \leq \exp(-C_{\mathrm{LS}} \gamma \ell / 2) \mathsf{KL}(\nu \,\|\, \pi) + 4\gamma d L^2 C_{\mathrm{LS}} .$$

Thus, the lower semi-continuity of the Kullback-Leibler for the weak topology (van Erven & Harremos, 2014, Theorem 19) and the convergence in total variation distance of (10) imply that

$$\mathsf{KL}(\pi_\gamma \,\|\, \pi) \leq \liminf_{\ell \to \infty} \mathsf{KL}\!\left( \nu \mathsf{R}_\gamma^\ell \,\|\, \pi \right) \leq 4\gamma d L^2 C_{\mathrm{LS}}$$

where $\nu$ is such that $\mathsf{KL}(\nu \,\|\, \pi) < \infty$. $\qquad \square$

### A.4. Proof of Theorem 3.3

Similarly to the sequence (9) in the proof of Proposition 3.2, define

$$\lambda_t^\star = \begin{cases} \dfrac{\log \int f_t^\varepsilon \mathrm{d}\mu_t K_\pi}{\log \int f_t^\varepsilon \mathrm{d}\mu_t K_\pi - \log \int f_t^\varepsilon \mathrm{d}\mu_t} & \text{if} \quad \log \int f_t^\varepsilon \mathrm{d}\mu_t K_\pi > 0, \\[2ex] \beta_t & \text{otherwise.} \end{cases} \tag{11}$$

Recall that $f_t$ is the Radon-Nikodym derivative of $\pi$ with respect to $\mu_t^{\mathsf{R}}$. The convexity of the Kullback-Leibler divergence and the definition of $\lambda_t^\star$ in (11) yield

$$\begin{aligned} \mathsf{KL}\!\left( \pi_\gamma \,\|\, \mu_{t+1}^{\mathsf{R}} \right) \leq{}& \lambda_t^\star \int \log \frac{\mathrm{d}\pi_\gamma}{\mathrm{d}f_t^\varepsilon \mu_t^{\mathsf{R}}} \mathrm{d}\pi_\gamma + (1 - \lambda_t^\star) \int \log \frac{\mathrm{d}\pi_\gamma}{\mathrm{d}f_t^\varepsilon \mu_t^{\mathsf{R}} \mathsf{R}_\gamma} \mathrm{d}\pi_\gamma \\ &+ \lambda_t^\star \log \int f_t^\varepsilon \mathrm{d}\mu_t^{\mathsf{R}} + (1 - \lambda_t^\star) \log \int f_t^\varepsilon \mathrm{d}\mu_t^{\mathsf{R}} \mathsf{R}_\gamma \\ \leq{}& \lambda_t^\star \int \log \frac{\mathrm{d}\pi_\gamma}{\mathrm{d}f_t^\varepsilon \mu_t^{\mathsf{R}}} \mathrm{d}\pi_\gamma + (1 - \lambda_t^\star) \int \log \frac{\mathrm{d}\pi_\gamma}{\mathrm{d}f_t^\varepsilon \mu_t^{\mathsf{R}} \mathsf{R}_\gamma} \mathrm{d}\pi_\gamma . \end{aligned}$$

Next,

$$\int \log \frac{\mathrm{d}\pi_\gamma}{\mathrm{d}f_t^\varepsilon \mu_t^{\mathsf{R}}} \mathrm{d}\pi_\gamma = (1 - \varepsilon) \mathsf{KL}\!\left( \pi_\gamma \,\|\, \mu_t^{\mathsf{R}} \right) + \varepsilon \mathsf{KL}(\pi_\gamma \,\|\, \pi) ,$$

$$\int \log \frac{\mathrm{d}\pi_\gamma}{\mathrm{d}f_t^\varepsilon \mu_t^{\mathsf{R}} \mathsf{R}_\gamma} \mathrm{d}\pi_\gamma \leq (1 - \varepsilon) \mathsf{KL}\!\left( \pi_\gamma \,\|\, \mu_t^{\mathsf{R}} \right) + \varepsilon \mathsf{KL}(\pi_\gamma \,\|\, \pi) ,$$

where the second inequality follows using the Data Processing inequality (van Erven & Harremos, 2014) and the fact that $\mathsf{R}_\gamma$ is $\pi_\gamma$-invariant. Thus, we have

$$\mathsf{KL}\!\left( \pi_\gamma \,\|\, \mu_{t+1}^{\mathsf{R}} \right) \leq (1 - \varepsilon) \mathsf{KL}\!\left( \pi_\gamma \,\|\, \mu_t^{\mathsf{R}} \right) + \varepsilon \mathsf{KL}(\pi_\gamma \,\|\, \pi) \leq (1 - \varepsilon)^t \mathsf{KL}\!\left( \pi_\gamma \,\|\, \mu_0^{\mathsf{R}} \right) + \mathsf{KL}(\pi_\gamma \,\|\, \pi)$$

and our upper bound is deduced from the Pinsker's inequality and Proposition A.1,

$$\begin{aligned} \left\| \pi - \mu_t^{\mathsf{R}} \right\|_{\mathsf{TV}} &\leq \left\| \pi - \pi_\gamma \right\|_{\mathsf{TV}} + \left\| \pi_\gamma - \mu_t^{\mathsf{R}} \right\|_{\mathsf{TV}} \\ &\leq (1 - \varepsilon)^{t/2} \sqrt{2 \mathsf{KL}\!\left( \pi_\gamma \,\|\, \mu_0^{\mathsf{R}} \right)} + 2\sqrt{2 \mathsf{KL}(\pi_\gamma \,\|\, \pi)} . \end{aligned}$$

The conclusion follows using Proposition A.1.

# B. Additional Elements on the Gaussian Mixture Benchmarks

This section reports additional results and implementation details for the Gaussian mixture benchmarks of Section 4.1.

## B.1. Additional Sliced Wasserstein Distances

Figure 6 illustrates the evolution of the sliced Wasserstein distance $\mathrm{SW}_2(\tilde{\mu}_t, \pi)$ across EM2C iterations $t$, and Table 4 reports final sliced Wasserstein distances $\mathrm{SW}_2(\tilde{\mu}_T, \pi)$ for the various target distributions. Across all configurations, they confirm the trends observed in the main paper: smaller values of the mixing parameter $\lambda$ lead to improved approximation of the target distribution.

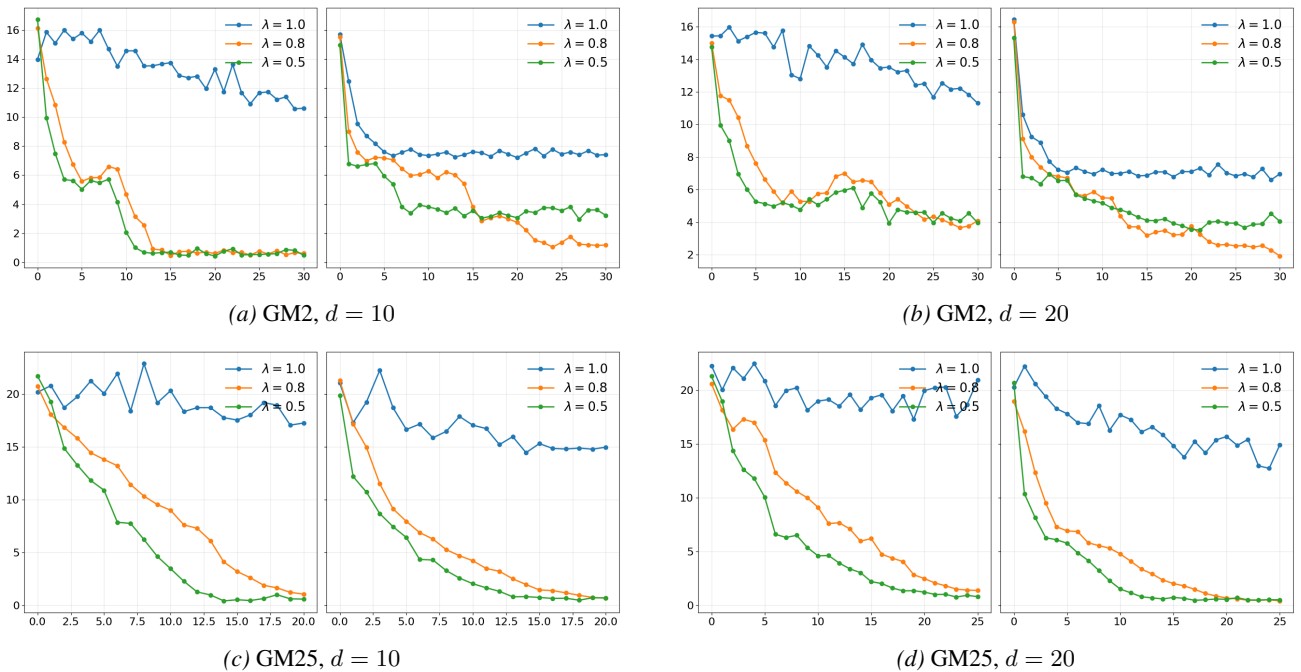

*(a) GM2, $d = 10$*    *(b) GM2, $d = 20$*

*(c) GM25, $d = 10$*    *(d) GM25, $d = 20$*

*Figure 6.* Evolution of the sliced Wasserstein distance $\mathrm{SW}_2(\tilde{\mu}_t, \pi)$ over EM2C iterations for additional Gaussian mixture benchmarks. Results are shown for Random Walk (left) and Unadjusted Langevin (right) exploration kernels, and for $\lambda \in \{0.5, 0.8, 1.0\}$.

*Table 4.* Final sliced Wasserstein distances of EM2C for Gaussian mixture benchmarks GM2, GM4, and GM25 at various ambient dimensions using Random Walk (RW) or Unadjusted Langevin Algorithm (ULA) exploration kernels.

| | | GM2 | | GM4 | | GM25 | |
|---|---|---|---|---|---|---|---|
| $d$ | $\lambda$ | RW | ULA | RW | ULA | RW | ULA |
| 4 | 1.0 | 7.66 | 7.44 | 17.98 | 18.14 | 12.10 | 11.43 |
| 4 | 0.8 | 7.37 | 2.31 | – | – | 2.63 | 0.60 |
| 4 | 0.5 | 4.84 | 4.30 | – | – | 0.88 | 0.36 |
| 10 | 1.0 | 7.31 | 7.20 | 18.71 | 16.39 | – | – |
| 10 | 0.8 | 0.62 | 1.19 | 7.10 | 0.63 | 1.08 | 0.70 |
| 10 | 0.5 | 0.49 | 3.23 | 0.79 | 0.90 | 0.61 | 0.72 |
| 20 | 1.0 | 6.84 | 7.59 | – | – | – | – |
| 20 | 0.8 | 3.83 | 1.59 | 6.35 | 5.74 | 1.39 | 0.42 |
| 20 | 0.5 | 3.19 | 4.01 | 1.64 | 3.95 | 0.81 | 0.52 |

## B.2. Additional Qualitative Evolution Plots

Figures 7–9 complement Figure 3. They illustrate the effect of the mixing parameter $\lambda$ and the exploration kernel on the ability of EM2C to recover and maintain multimodal structure.

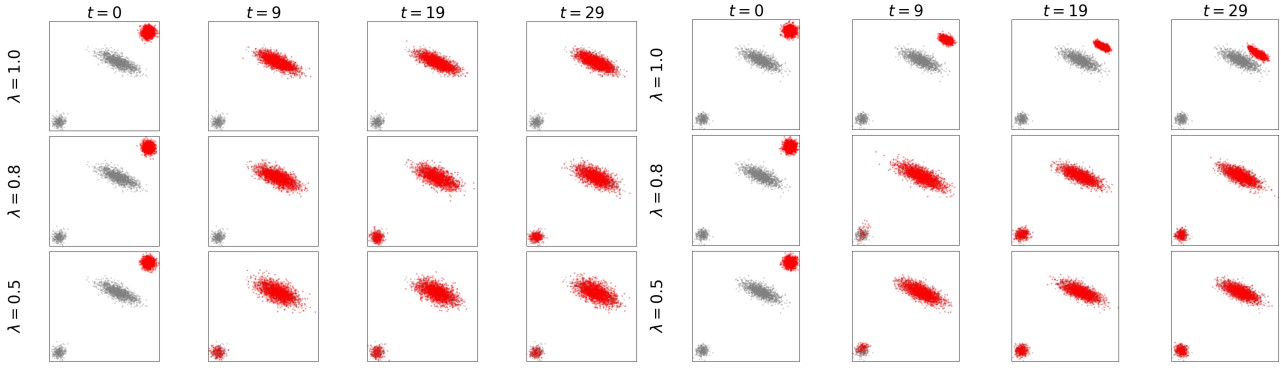

*(a)* Unadjusted Langevin exploration kernel      *(b)* Random Walk Metropolis–Hastings exploration kernel

*Figure 7.* Evolution of EM2C proposals for the GM2 target ($d = 10$). Rows correspond to $\lambda \in \{0.5, 0.8, 1.0\}$ and columns to EM2C iterations $t$. Gray and red points are samples from the target distribution $\pi$ and the learned proposal $\tilde{\mu}_t$, respectively. The marginal samples on the first two coordinates are displayed.

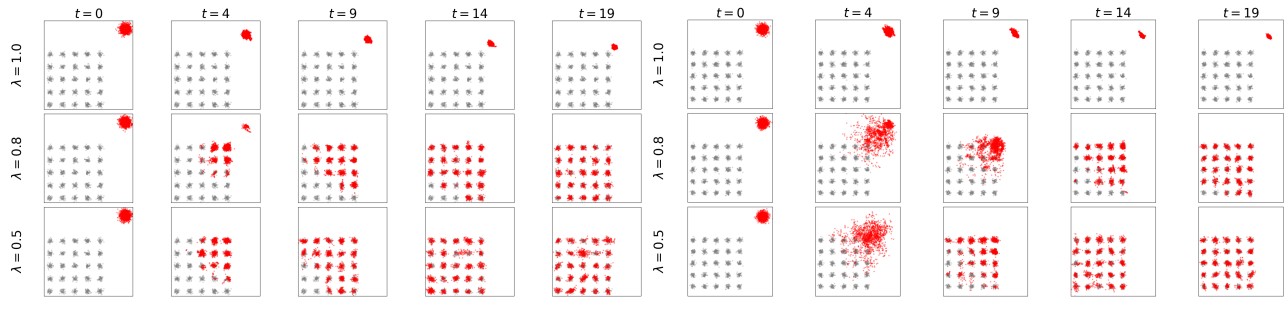

*(a)* Unadjusted Langevin exploration kernel      *(b)* Random Walk Metropolis-Hastings exploration kernel

*Figure 8.* Evolution of EM2C proposals for the GM25 target ($d = 10$). Rows correspond to $\lambda \in \{0.5, 0.8, 1.0\}$ and columns to EM2C iterations $t$. Gray and red points are samples from the target distribution $\pi$ and the learned proposal $\tilde{\mu}_t$, respectively. The marginal samples on the first two coordinates are displayed.

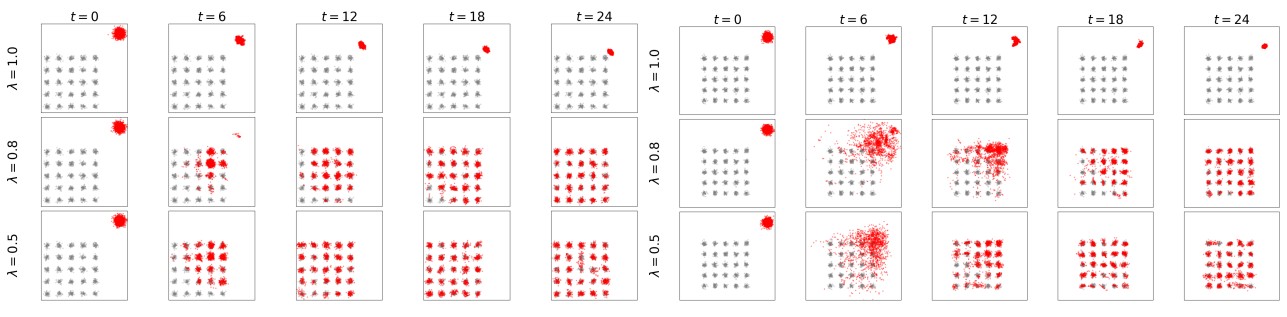

*(a)* Unadjusted Langevin exploration kernel      *(b)* Random Walk Metropolis–Hastings exploration kernel

*Figure 9.* Evolution of EM2C proposals for the GM25 target ($d = 20$). Rows correspond to $\lambda \in \{0.5, 0.8, 1.0\}$ and columns to EM2C iterations $t$. Gray and red points are samples from the target distribution $\pi$ and the learned proposal $\tilde{\mu}_t$, respectively. The marginal samples on the first two coordinates are displayed.

## B.3. Comparison with Markov Chain Monte Carlo

Figure 10 shows a representative run of ULA on GM4 in dimension $d = 10$, providing a direct qualitative comparison with the EM2C proposal evolution shown in Figure 3. This ULA baseline is run under the matched computational budget described in Appendix E.5, using the hyperparameter $\gamma_K = 2.0$. It corresponds to the limiting case $\lambda = 0$, where only the Markov kernel is used, and is preferred to RW because it performed better on this benchmark. In contrast to EM2C, which maintains a population of particles, the single-chain MCMC sampler exhibits slower mode exploration and delayed coverage of the target distribution.

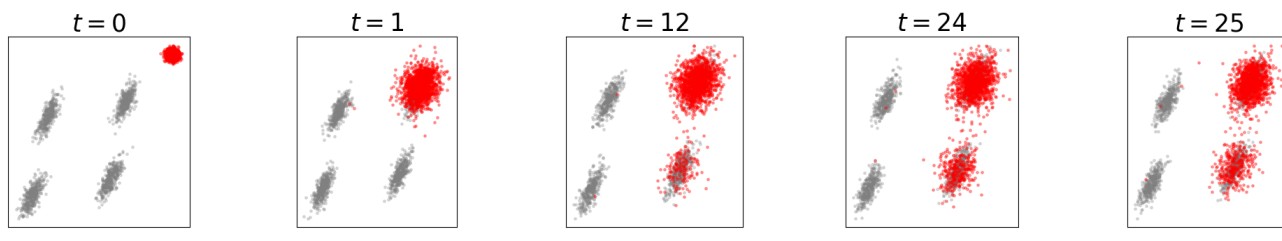

*Figure 10.* Evolution of the Unadjusted Langevin Algorithm (ULA) for the GM4 target distribution ($d = 10$). Columns corresponds to ULA iterations $t$. Gray points denote samples from the target distribution $\pi$, and red points correspond to the current state of the Markov chain. The marginal samples on the first two coordinates are displayed.

## B.4. Experimental Design

**Exploration kernel $K$.** Table 5 reports the kernel parameters (RW: step size $\sigma_K$, ULA: step size $\gamma_K$), number of kernel steps $n_K$ applied at each EM2C iteration, and number of EM2C iterations $T$ used in the Gaussian mixture benchmarks. These configurations are adapted to the scale and geometry of each targets and were selected to ensure stable exploration across dimensions.

*Table 5.* Exploration kernel hyperparameters and number of EM2C iterations for the Gaussian mixture benchmarks.

| | | GM2 | | | GM4 | | | GM25 | | |
|---|---|---|---|---|---|---|---|---|---|---|
| $d$ | Kernel | $\sigma_K/\gamma_K$ | $n_K$ | $T$ | $\sigma_K/\gamma_K$ | $n_K$ | $T$ | $\sigma_K/\gamma_K$ | $n_K$ | $T$ |
| $2,4$ | RW | 6.0 | 20 | 25 | 4.5 | 15 | 25 | 1.5 | 10 | 15 |
| 10 | RW | 8.0 | 20 | 30 | 4.5 | 15 | 25 | 2.0 | 15 | 20 |
| 20 | RW | 7.0 | 20 | 30 | 5.0 | 15 | 25 | 2.5 | 20 | 25 |
| $2,4$ | ULA | 2.3 | 15 | 25 | 2.0 | 10 | 25 | 0.3 | 10 | 15 |
| 10 | ULA | 2.3 | 15 | 30 | 2.0 | 10 | 25 | 0.3 | 10 | 20 |
| 20 | ULA | 2.3 | 15 | 30 | 2.0 | 10 | 25 | 0.35 | 10 | 25 |

**Variational family.** For Gaussian mixture benchmarks, $\tilde{\mu}_t$ belongs to a tensorized Gaussian mixture family. Writing $x = (x^{(1)}, \ldots, x^{(d/2)})$ with $x^{(j)} \in \mathbb{R}^2$,

$$\mu_\theta(x) = \prod_{j=1}^{d/2} \mu_\theta^{(j)}(x^{(j)}), \qquad \mu_\theta^{(j)}(x) = \sum_{k=1}^{K_0} w_k^{(j)} \mathcal{N}(x \mid m_k^{(j)}, \Sigma_k^{(j)}) \,.$$

The covariance matrices are full and $K_0$ matches the number of modes of the two-dimensional marginal: $K_0 = 2, 4, 25$ for GM2, GM4, and GM25, respectively. This tensorized construction implicitly defines a mixture with $K_0^{d/2}$ components in dimension $d$.

**Projection via EM algorithm.** At each EM2C iteration, the oracle update eq:memd:iterates is approximated by the empirical measure of particles $Z_t^{1:N}$. For each block $j$, the two-dimensional GMM $\mu_\theta^{(j)}$ is fitted by maximum likelihood to $Z_{t,j}^{1:N}$ using EM. We use full covariance matrices, k-means++ initialization, $n_{\text{init}} = 3$ random restarts, at most $500$ EM iterations, and covariance regularization $10^{-3}$. All EM fits are performed independently across blocks and EM2C iterations.

### B.5. High-Dimensional Gaussian Mixture

**Target distribution.** We consider a Gaussian mixture on $\mathbb{R}^d$, $d = 200$, with $K = 4$ components, designed to isolate ambient dimensionality from combinatorial multimodality. It follows high-dimensional sampling benchmarks from Chen et al. (2024). The target distribution is defined as

$$\pi(x) = \sum_{k=1}^{4} w_k \, \mathcal{N}(x \mid m_k, 0.05 \, I_d), \quad x \in \mathbb{R}^d.$$

where $w = (0.1, 0.1, 0.1, 0.7)$ and $m_k = (m_k^{(2D)}, 0, \ldots, 0)$ with

$$m_k^{(2D)} \in \{(-1, 1), (1, 1), (-1, -1), (1, -1)\}.$$

**Experimental setup.** The initial proposal is $\mu_0 = \mathcal{N}(-2\mathbf{1}_d, 0.1^2 I_d)$, which has little overlap with the target to create a difficult initialization regime and tests the robustness of the algorithm. We run EM2C with $T = 15$, $N = 8,000$, $\varepsilon = 0.8$, $\lambda = 0.9$, and ULA kernel with parameters $\gamma_K = 0.02$, $n_K = 15$. At each iteration, the projection fits a Gaussian mixture with diagonal-covariance

$$\mu_\theta(x) = \sum_{k=1}^{10} w_k \, \mathcal{N}(x \mid m_k, \Sigma_k)$$

by EM, using covariance regularization $10^{-4}$ and a maximum of 300 EM iterations. It is implemented using the scikit-learn version of EM with 3 random initializations.

**Discussion.** Figure 11 shows the sliced Wasserstein trajectory (a), and and final samples projected onto the first two coordinates (b). EM2C recovers the four-mode structure despite the high ambient dimension and challenging initialization.

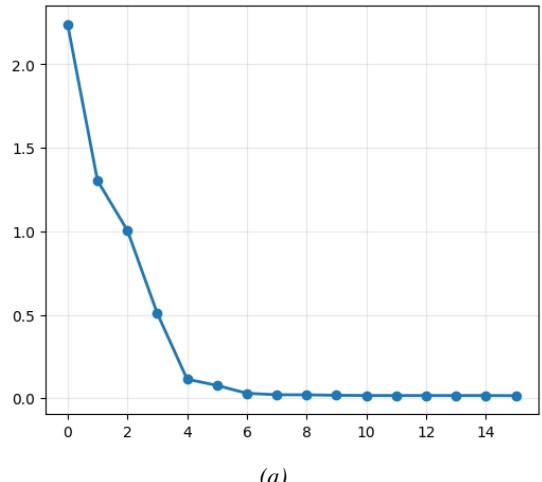
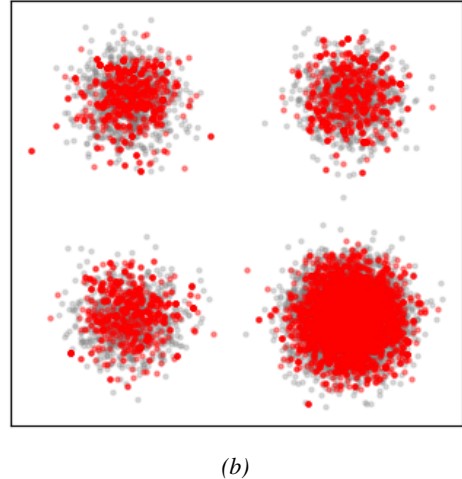

*(a)*          *(b)*

*Figure 11.* (a) Evolution of the sliced Wasserstein distance $\mathrm{SW}_2(\tilde{\mu}_t, \pi)$ over EM2C iterations ($d = 200$). (b) Final samples obtained by EM2C. Gray and red points are samples from the target distribution $\pi$ and the learned proposal $\tilde{\mu}_t$, respectively. The marginal samples on the first two coordinates are displayed.

## C. Additional Elements on the Dual Moons and Two Rings Benchmarks

This section gives additional results and implementation details for the two-dimensional benchmarks of Section 4.2.

### C.1. Additional Qualitative Evolution Plots

Figures 12–13 show EM2C proposal evolution for $\lambda \in \{0.8, 1.0\}$. For visualization pupeses only, samples from $\tilde{\mu}_t$ are resampled with weights proportional to $\pi(x)/\tilde{\mu}_t(x)$, removing particles with negligible target contribution.

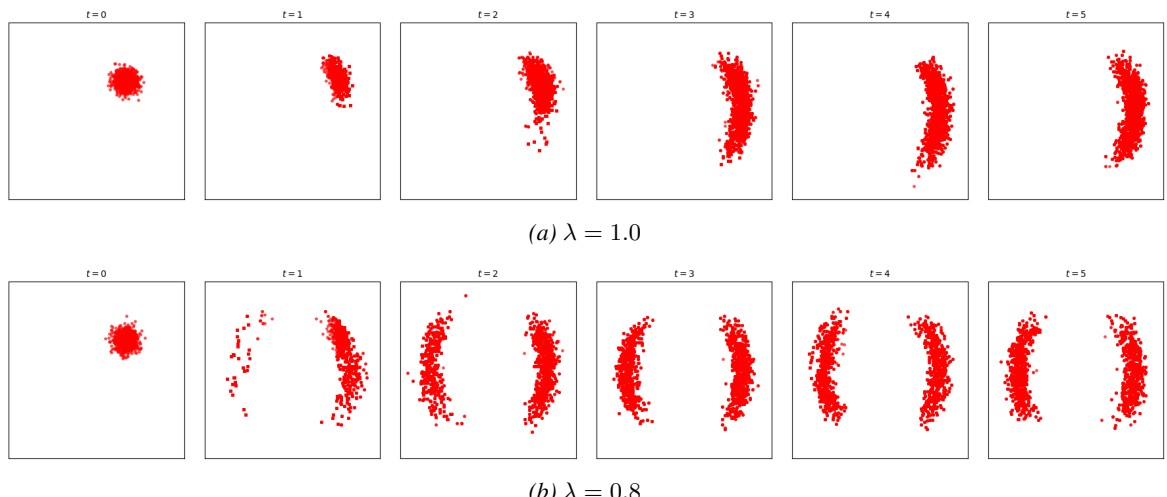

*Figure 12.* Evolution of EM2C proposal distributions on the Dual Moons benchmark. Plotted samples are drawn from the intermediate proposals $\tilde{\mu}_t$ and resampled using importance weights $\pi(x)/\tilde{\mu}_t(x)$.

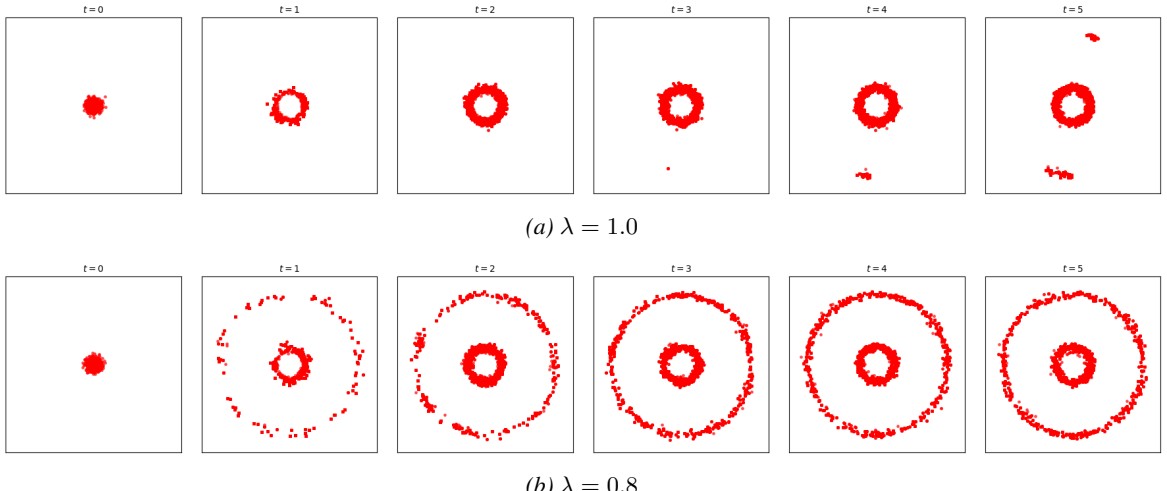

*Figure 13.* Evolution of EM2C proposal distributions on the Two Rings benchmark. Plotted samples are drawn from the intermediate proposals $\tilde{\mu}_t$ and resampled using importance weights $\pi(x)/\tilde{\mu}_t(x)$.

### C.2. Experimental Design

We report here the initial distributions and the main hyperparameters associated with each algorithm used throughout experiments of Section 4.2.

**Initial distributions**   For all methods, the initial distribution is set to a Gaussian, $\mathcal{N}(m_0, \Sigma_0)$. For the Dual Moons benchmark, $m_0 = (1,1)^\top$ and $\Sigma_0 = 0.04 I_2$. For the Two Rings benchmark, $m_0 = (0,0)^\top$ and $\Sigma_0 = 0.04 I_2$.

**Methods and hyperparameters**   All methods use the same initialization and $N = 10{,}000$ particles. Hyperparameters are chosen to give comparable target-density and gradient evaluation budgets, of the order of $10^7$ evaluations. The cost model is given in Appendix E.5. Table 6 summarizes the main parameters of the following methods.

#### C.2.1. EM2C CONFIGURATION

Algorithm 1 is run for $T = 6$ iterations, $\varepsilon = 0.8$, $\lambda = 0.8$. $K$ is set to a RW kernel applied for $n_K$ steps with Gaussian proposal $\mathcal{N}(0, \sigma_K^2)$. Before the projection step, we perform a local resample-move step (see Appendix E.4 for details) using

the same RW kernel applied for $n_L$ steps with Gaussian proposal $\mathcal{N}(0, \sigma_L^2)$.

The projection step uses a Neural Spline Flow. The flow has three coupling transforms, each parameterized by a two-hidden-layer neural network of width 64, with, for each coupling layer, rational quadratic splines transformations using 8 bins. It is trained by maximum likelihood for 8 epochs on $N = 10,000$ particles, with batch size 256 and learning rate $10^{-3}$. The architecture and hyperparameters of the flow are kept fixed across target distributions and experiments.

### C.2.2. COMPETING METHODS CONFIGURATIONS

**RW:** Metropolis–Hastings sampler run as $N$ independent chains of length $n_{\text{iter}}$, using Gaussian proposal $\mathcal{N}(0, \sigma_{\text{RW}}^2 I_2)$ and retaining only the final state of each chain. We do not include ULA for these experiments, as gradient-based proposals were empirically less effective than RW in this low-dimensional and multimodal setting.

**NUTS:** hyperparameters of NUTS are tuned during a warmup phase of $n_{\text{warmup}}$ iterations, with $0.8$ target acceptance probability. The maximum tree depth is set to $d_{\max}$. After warmup, $N$ samples are collected from the resulting chain.

**AIS:** the method uses a linear annealing schedule $\{\beta_\ell\}_{\ell=0}^L$. At each intermediate temperature, particles are first reweighted and then propagated using $n_{\text{iter}}$ steps of a RW kernel targeting the intermediate distribution $\pi_{\beta_\ell}$. The proposal variance $\sigma_{\text{RW}}^2$ is tuned independently from the MCMC baseline to ensure efficient exploration across intermediate distributions while maintaining a computational budget comparable to the other methods.

**DiGS:** the method alternates between (i) Gaussian perturbation steps and (ii) local Langevin dynamics targeting a denoising posterior. Given a current state $x$, it generates a noisy variable

$$\tilde{x} = \alpha x + \varepsilon \sqrt{1 - \alpha^2}, \qquad \varepsilon \sim \mathcal{N}(0, I),$$

Conditionally on $\tilde{x}$, Langevin-based updates are then performed to sample from the associated denoising distribution.

We follow the reference implementation of Chen et al. (2024). We run $n_{\text{chains}}$ parallel DiGS chains of $\lfloor N/n_{\text{chains}} \rfloor$ iterations, recording the state of each chain after every iteration. Each DiGS iteration provides a recorded sample and is controlled by: (i) a Langevin step size $\gamma$, (ii) a number of Gibbs sweeps $n_{\text{sweeps}}$, (iii) a number of Langevin steps per sweep $n_{\text{LD}}$, and (iv) a variance-preserving annealing schedule $(\alpha_{\min}, \alpha_{\max}, T_\alpha)$.

*Table 6.* Sampling methods hyperparameters for the Dual Moons and Two Rings benchmarks.

| Method | Parameters | Dual Moons | Two Rings | Comments |
|---|---|---|---|---|
| EM2C | $\sigma_K$, $n_K$, $\sigma_L$, $n_L$ | 1.0, 10, 0.1, 5 | 0.9, 10, 0.1, 5 | RW exploration and local move |
| RW | $\sigma_{\text{RW}}$, $n_{\text{iter}}$ | 1.0, 1,000 | 0.9, 1,000 | Shared RW kernel |
| NUTS | $d_{\max}$, $n_{\text{warmup}}$ | 9, 1,000 | 9, 1,000 | Single chain, adaptive HMC parameters |
| AIS | $L$, $\sigma_{\text{RW}}$, $n_{\text{iter}}$ | 10, 2.0, 100 | 10, 2.0, 100 | Larger variance, fewer steps (budget-matched) |
| DiGS | $n_{\text{chains}}$, $\gamma$, $n_{\text{sweeps}}$, $n_{\text{LD}}$ | 50, 0.05, 50, 2 | 50, 0.05, 50, 2 | Budget-matched Langevin updates |
| | $\alpha_{\min}$, $\alpha_{\max}$, $T_\alpha$ | 0.1, 0.9, 4 | 0.1, 0.9, 4 | |

## D. Additional Elements on the Bayesian Neural Network Benchmark

This section reports the main hyperparameters used in the Bayesian neural network experiments of Section 4.3.

**Methods and hyperparameters.** All methods are initialized from the prior and generate $N = 500$ samples for evaluation. Hyperparameters are chosen to give comparable target-density and gradient evaluation budgets. The cost model is given in Appendix E.5. MALA and DiGS use an evaluation budget of approximately $10^7$, while EM2C uses an evaluation budget of about $8 \times 10^6$ to account for projection costs.

### D.1. EM2C configuration

Algorithm 1 is run for $T = 750$ iterations, $\varepsilon = 0.8$, $\lambda = 0.5$. $K$ is set to a MALA kernel applied for $n_K = 20$ steps with step size $\gamma = 10^{-4}$. Before the projection step, we perform a local resample-move step (see Appendix E.4 for details) using the same MALA kernel applied for $n_L = 2$ steps.

At each iteration, the projection fits a Gaussian mixture with diagonal-covariance

$$\mu_\theta(x) = \sum_{k=1}^{20} w_k \, \mathcal{N}(x \mid m_k, \Sigma_k)$$

by EM, using covariance regularization $10^{-4}$ and a maximum of 100 EM iterations.

### D.2. Competing Methods Configurations

**MALA:** we run $n_{\text{chains}} = 8$ parallel chains with step size $\gamma = 10^{-4}$ and record samples every 10,000 MALA steps.

**DiGS:** we use the same implementation as in Appendix C.2 with $n_{\text{chains}} = 8$ parallel chains, $T_\alpha = 8$ noise levels linearly interpolating from $\alpha_T = 0.1$ to $\alpha_1 = 0.9$, $n_{\text{sweeps}} = 200$, $n_{\text{LD}} = 10$ MALA steps per sweep, and step size $\gamma = 10^{-4}$.

## E. Additional Implementation Details

This section summarizes the evaluation metrics, the optional local move used in some experiments, and the cost model used for budget matching.

### E.1. Sliced Wasserstein Distance

We use the sliced Wasserstein distance to compare proposal and target samples. This metric is computationally tractable in high dimension, while retaining sensitivity to geometric discrepancies between distributions.

**Definition E.1.** Let $\mu$ and $\nu$ be two probability measures on $\mathbb{R}^d$ with finite second moments. For a unit vector $\theta \in \mathbb{S}^{d-1}$, denote by $\theta_\# \mu$ the pushforward of $\mu$ by $x \mapsto \langle x, \theta \rangle$. The sliced Wasserstein-2 distance between $\mu$ and $\nu$ is defined as

$$\text{SW}_2(\mu, \nu) = \left( \int_{\mathbb{S}^{d-1}} W_2^2(\theta_\# \mu, \theta_\# \nu) \, \mathrm{d}\theta \right)^{1/2},$$

where $W_2$ is the one-dimensional Wasserstein-2 distance and the integral is with respect to the uniform distribution on $\mathbb{S}^{d-1}$.

**Empirical computation.** Given samples $\{x_i\}_{i=1}^N \sim \mu$ and $\{y_i\}_{i=1}^N \sim \nu$, we approximate the integral using directions $\{\theta_k\}_{k=1}^K$ sampled uniformly in $\mathbb{S}^{d-1}$, and projections $\langle x_i, \theta_k \rangle$ and $\langle y_i, \theta_k \rangle$. Namely we use the Monte Carlo estimator

$$\widehat{\text{SW}}_2(\mu, \nu) = \left( \frac{1}{K} \sum_{k=1}^K \frac{1}{N} \sum_{i=1}^N \left( x_{(i)}^{(k)} - y_{(i)}^{(k)} \right)^2 \right)^{1/2},$$

where $x_{(i)}^{(k)}$ and $y_{(i)}^{(k)}$ denote the sorted projected samples along direction $\theta_k$.

**Implementation details.** For each benchmark, $\widehat{\text{SW}}_2$ is computed using the $N$ samples produced by each method, a fixed reference set of $N$ target samples shared across methods, and $K = 100$ random projection directions. For EM2C, the same target samples are also used across iterations.

### E.2. Energy Distance

For the two-dimensional benchmarks of Section 4.2, we also report the energy distance which complements $\text{SW}_2$ by capturing discrepancies in both location and spread.

**Definition E.2.** Let $X \sim \mu$ and $Y \sim \nu$ be random variables in $\mathbb{R}^d$, and let $X'$ and $Y'$ be independent copies of $X$ and $Y$, respectively. The energy distance between $\mu$ and $\nu$ is defined as

$$\text{ED}(\mu, \nu) = 2\, \mathbb{E}[\|X - Y\|] - \mathbb{E}[\|X - X'\|] - \mathbb{E}[\|Y - Y'\|] \, .$$

**Empirical computation.** Given samples $\{x_i\}_{i=1}^N \sim \mu$ and $\{y_j\}_{j=1}^N \sim \nu$, we use the estimator

$$\widehat{\mathrm{ED}}(\mu, \nu) = \frac{2}{N^2} \sum_{i=1}^N \sum_{j=1}^N \|x_i - y_j\| - \frac{1}{N^2} \sum_{i=1}^N \sum_{i'=1}^N \|x_i - x_{i'}\| - \frac{1}{N^2} \sum_{j=1}^N \sum_{j'=1}^N \|y_j - y_{j'}\|.$$

**Implementation details.** The energy distance $\widehat{\mathrm{ED}}$ is computed using $N = 10{,}000$ samples drawn from both the proposal distribution $\tilde{\mu}_t$ and target distribution $\pi$.

### E.3. Test Negative Log-Likelihood

For Bayesian neural network benchmark of Section 4.3, we evaluate posterior samples via negative log-likelihood (NLL) on a held-out test dataset.

**Definition E.3.** Let $\mathcal{D}_{\mathrm{test}} = \{(x_i, y_i)\}_{i=1}^{N_{\mathrm{test}}}$ be a test dataset. For a distribution $\mu$ over network parameters, define

$$\mathrm{NLL} = \mathbb{E}_{\theta \sim \mu}\left[ -\frac{1}{N_{\mathrm{test}}} \sum_{i=1}^{N_{\mathrm{test}}} \log p(y_i \mid x_i, \theta) \right]. \tag{12}$$

In our setting, $p(y \mid x, \theta) = \mathcal{N}(f_\theta(x), \sigma_n^2)$. Thus each term in (12) corresponds to the usual squared-error loss with Gaussian noise.

**Empirical computation.** Given samples $\{\theta_j\}_{j=1}^N$ from the posterior approximation produced by each method, we estimate

$$\widehat{\mathrm{NLL}} = \frac{1}{N} \sum_{j=1}^N \left[ -\frac{1}{N_{\mathrm{test}}} \sum_{i=1}^{N_{\mathrm{test}}} \log p(y_i \mid x_i, \theta_j) \right].$$

For instance, the posterior approximation from EM2C is the learned proposal $\tilde{\mu}_T$.

**Implementation details.** In all experiments, we use the $N = 500$ posterior samples for evaluation and a test dataset of size $N_{\mathrm{test}} = 500$.

Since EM2C updates a population of particles, its NLL can fluctuate more across iterations than for chain-based samplers. To reduce this variability, we report the average NLL over the last 20 iterations of each run.

### E.4. Optional Local Move Kernel

In Algorithm 1, the particles $Z_t^{1:N}$ are resampled from the empirical mixture $\widehat{\mathcal{F}}_{\tilde{\mu}_t}^N$, which may result in reduced diversity among the resampled particles. In some experiments, after this resampling step, we apply a short local Markov move, that is, for a Markov kernel $L$, independently for each particle, we compute

$$\widetilde{Z}_t^i \sim L(Z_t^i, \cdot), \qquad i = 1, \ldots, N,$$

and use $\widetilde{Z}_t^{1:N}$ in the projection step. This optional move mildly disperses particles and improves numerical stability. It is omitted from Algorithm 1 in the main text for clarity, as it is a standard diversity-enhancing refinement and does not change the logic of the Monte Carlo implementation.

### E.5. Computational Budget Rules

We compare methods using approximate evaluation budgets, counted in units of target-density or gradient evaluations. For gradient-based methods, one gradient evaluation is counted as one evaluation unit; for Metropolis-corrected methods such as MALA and NUTS, we include the additional density evaluations required by the acceptance step. For NUTS, we use $2^{d_{\max}}$ as an upper bound on the number of leapfrog steps per iteration, since tree expansion may stop earlier due to the U-turn criterion. The evaluation budget for the different methods used throughout the paper is reported in Table 7

Proposal-density evaluations, sampling, resampling, and projection costs are not included in the budget. To compensate for the additional cost of projection, especially when training normalizing flows, EM2C is run with a slightly reduced evaluation

budget. Moreover, when $\lambda = 1$, the exploration kernel is not used within Algorithm 1. To provide comparison under a matched budget, we reallocate the corresponding kernel budget to $n_K$ additional EM2C iterations.

These budgets are used only to choose comparable hyperparameters, not to provide wall-clock comparisons.

*Table 7.* Approximate number of target or gradient evaluations used for budget matching.

| Method | Evaluation budget |
| --- | --- |
| EM2C with RW or ULA exploration kernel | $N\,T\,(n_K + n_L + 2)$ |
| EM2C with MALA exploration kernel | $N\,T\,(2n_K + 2n_L + 2)$ |
| RW | $n_{\text{iter}}\,n_{\text{chains}}$ |
| ULA | $n_{\text{iter}}\,n_{\text{chains}}$ |
| MALA | $2\,n_{\text{iter}}\,n_{\text{chains}}$ |
| NUTS | $\lesssim 2^{d_{\max}+1}\,(n_{\text{warmup}} + N)$ |
| DiGS | $2N\,T_\alpha\,n_{\text{sweeps}}\,(n_{\text{LD}} + 1)$ |
| AIS | $N\,L\,(n_{\text{iter}} + 1)$ |

