# OpenReview forum: "Entropic Mirror Monte Carlo"
_ICML.cc/2026/Conference — ICML 2026 regular_

### Official Review · Reviewer_Lwf3 · 2026-02-27

**Soundness:** 3
**Presentation:** 3
**Significance:** 3
**Originality:** 3
**Overall Recommendation:** 5
**Confidence:** 3

**Summary:**

This paper proposes Entropic Mirror Monte Carlo (EM2C), a distribution-level iterative sampling framework combining an entropic mirror mapping with a Markov kernel. The authors include theoretical convergence analysis (Prop 3.2 and Theorem 3.3) and an implentation in Algorithm 1.

**Compliance With Llm Reviewing Policy:**

Affirmed.

**Key Questions For Authors:**

1. The bounded density assumption $\||d\pi/d\mu_0\||< \infty$ appears important, but I wonder how realistic this is in high-dimensions. Even, if in principle, the derivative is bounded, the required sample size for stable weights becomes very large. Can you address this more carefully?
2. A method like particle mirror descent (Dai et al. AISTATS 2016), which is cited, does not require this global bounded density ratio, can you speak more about the pros and cons of this method among other mirror descent strategies?
3. I'm interested in the projection step in Algorithm 1. It seems that the KL contraction guarantees in Prop 3.2 and Thm 3.3 apply to the idealised mapping. However, Algorithm 1 introduces a Monte Carlo approximation followed by a projection onto a parametric family $\mathcal{F}_\theta$. Is there any theoretical control of the error introduced by this projection step?
4. Also on the projection step, how sensitive is the method to misspecification? For example, if $\mathcal{F}_\theta$ cannot represent multimodal structure accurately, can the KL mechanism still operate effectively?

**Limitations:**

The theoretical guarantees rely on some assumptions, including bounded density ratios, which may be unrealistic in high-dimensional or highly multimodal settings. Moreover, the KL contraction results apply to the idealised mapping, while the practical algorithm introduces Monte Carlo approximation and projection onto a parametric family, for which no theoretical error control is provided.

**Strengths And Weaknesses:**

Strengths: The paper is well written in general and provides a principled construction of a contracting sequence of proposal distributions for IS. The KL contraction analysis is technically sound and clearly presented. The approach is conceptually elegant and bridges adaptive IS and MCMC in a unified framework.

Weaknesses: The theoretical guarantees rely on assumptions which may not hold in challenging high-dimensional or highly multimodal settings. My main concerns is the defining of the projective parametric family in the implementation of EM2C (Algorithm 1). The convergence analysis applies to the idealised mapping only, while the practical algorithm introduces Monte Carlo approximation and projection onto a parametric family without theoretical control of projection error. Please see 'Key Questions for Authors' for further comments and requests for clarification.

---

> ### Author Rebuttal · Authors · 2026-03-30
>
> We thank the reviewer for the very positive and thoughtful assessment of our work. We particularly appreciate the recognition of the conceptual contribution in bridging adaptive importance sampling and MCMC, as well as the careful identification of some limitations. We fully agree that clarifying these aspects will strengthen the paper.
>
> 1. Bounded density ratio assumption can be restrictive, especially in high-dimensional or highly multimodal settings. Its role in our analysis is to ensure that the mirror-descent update is well-defined and that the initial discrepancy between $\pi$ and $\mu_0$ is controlled. However, as the reviewer points out, even when the ratio is bounded, the effective sample size may still be poor in practice. This observation precisely motivates our approach: combining mirror reweighting with kernel-based exploration aims to progressively improve coverage of the target and alleviate weight degeneracy. We added a point to precise both the role and limitations of this assumption.
>
> 2. The key difference is that our update explicitly relies on the density ratio which requires some control on $\frac{d\pi}{d\mu_t}$. In contrast, particle mirror descent methods typically rely on gradient-based transport updates involving $\nabla \log \pi$ and $\nabla \log \mu$, and therefore do not require global boundedness of the density ratio.
>
> 3. Our theory studies an idealized contractive map that cannot be implemented. The role of $\mathsf{F}_\theta$ is not covered by the theory: it is introduced to obtain a tractable proposal $(\tilde\mu_t)$ that can be sampled from and evaluated. We agree that the current version does not sufficiently separate these two levels. In the revision, a complete paragraph clarifies (i) the theoretical results obtained for the "oracle" algorithm  and (ii) this projection is a practical approximation layer the practical Monte Carlo and optimization steps used to provide an efficient algorithm. Although these steps are not covered by the theory several directions for future works are discussed to control these approximations. Extending the analysis to include projection error is a major direction for future work.
>
> 4. The performance of EM2C depends on the expressiveness of the chosen family $\mathsf{F}_\theta$. In particular, if the family cannot represent multimodal structures, the projection step may collapse multiple modes into a simpler approximation. However, the weighting mechanism still provides insight on the mismatch between $\pi$ and the current proposal. In practice, the algorithm may still identify important regions of the space, but the projection step may limit how well these regions are jointly represented. We now mention this in the revised paper.
>
> **Additional remark.**
> Our mehtod outperforms some recent methods in challenging settings (especially multimodality) still commonly used in recent works even if the dimension remains small. However, to follow other reviewer's suggestions, we strengthened the empirical section by including a recent diffusion-based baseline (DiGS, Chen et al., 2024), and added an experimental setting in higher dimension ($d=200$). EM2C also outperforms DiGS. See reviewer JC2J for details.
>
> We believe the clarifications and additions described above, addresses the reviewer’s concerns and further strengthens the paper.

---

> > ### Author Rebuttal · Reviewer_Lwf3 · 2026-04-02
> >
> > Thank you to the authors for the comments and clarifications. I believe these additions will improve the readability of the paper. My original score stands.

---

> > > ### Author Response · Authors · 2026-04-03
> > >
> > > Thank you for this positive and encouraging follow-up. We are pleased that the additions, following your comments, improve the readability of the paper.

---

### Official Review · Reviewer_BwJ5 · 2026-03-09

**Soundness:** 3
**Presentation:** 3
**Significance:** 3
**Originality:** 3
**Overall Recommendation:** 5
**Confidence:** 4

**Summary:**

The paper investigates a new method, called entropic Monte Carlo, which goal is sampling from a target distribution. The method relies on ideas developed recently in the importance sampling literature: instead of using standard importance weights the authors propose to use tempered weights as is recommended by the miror descent algorithm. The approach developed is also related to (unajusted) Langevin Monte Carlo sampling as (unajusted) Langevin steps are used to draw sample from the proposition. In the end the proposal distribution, is a mixture between importance sampling steps with mirror descent weights and Langevin steps. A theory is given to show that iterating the steps of the algorithms lead to a convergence in distribution with respect to KL (Th. 3.2) and TV (Theorem 3.3). The theory is illustrated with some simulation including targets of multimodal Gaussians (in dimension 10 and 20) and other target with more complexe geometry (''two rings'' and ''dual moons'' in dimension 2).

**Compliance With Llm Reviewing Policy:**

Affirmed.

**Final Justification:**

My concerns have been addressed by the authors rebuttal. As a consequence, I increased my score by 1 points.

**Key Questions For Authors:**

I encourage the authors to answer to some - not necessary all - of my above comments

**Limitations:**

yes

**Strengths And Weaknesses:**

The paper is well written and interesting. I particularly enjoyed the theoretical part, which is carefully developed. The results are appealing, and the proofs are clear, easy to follow, and fairly elementary, making the main arguments accessible. Here are a few suggestions that may help the authors to further improve their work:

0) The two theorems claim the existence of a certain sequence $(\lambda_t)$ to ensure a geometric upper bound. Nothing is said about the definition of this sequence and its behavior. Second, the authors may comment  about the impact of a wrong choice of $\lambda$ in their proof as well as in practice.

1) Knowledge of gradients should be further emphasized. This is not the case in standard importance sampling setup as usual importance sampling weights are defined without gradients. Regarding this, I believe that the evaluation of the methods in the simulation part is not that fair as some methods rely on the gradient and others don't.

2) There is a gap between theory and practice as the authors study an oracle algorithm that is far away from the one proposed in the paper. The first gap concerns the mirror descent step (2) that constitutes the main building block of the algorithm. To my knowledge, this step cannot be done in practice as one cannot obtain a particle with the exact distribution given in (2). The steps involving the Langevin kernel are clear to me. However, what still seems somewhat inconsistent is the use of the parametric family, which does not appear to be addressed by the theoretical analysis. The introduction of the variational family $F_\theta$ sounds like a good idea in practice but looks a bit confusing given the paper's focus and more importantly the paper's theory.


3) The authors use mirror descent based particles measure $\hat F_{\mu_t} ^N$ which might be seen as a particular SMC method (I therefore encourage the authors to carefully check whether such SMC-kernel based mutations have been considered or not in earlier work). A link with this recent paper would be interesting:
Bianchi, P., Delyon, B., Priser, V., & Portier, F. (2024). Stochastic mirror descent for nonparametric adaptive importance sampling. arXiv preprint arXiv:2409.13272.
This work proposes to re-weight new samples after one mutation step that is done using a local kernel step. The kernel that is used is different from that of the Langevin which makes the algorithm different from the one that is proposed by the authors. However the fact that there is (a) generation from the proposal with Mirror descent weights and (b) kernel steps that are conducted makes the two algorithms related. In the above paper, the Kernel is just a Gaussian so the gradient term in the Langevin's auther step is removed.  Also after reading the Appendix, I understood that a local move $L$ is performed additionally but less is said about $L$ (if $L$ is a Gaussian kernel, then this makes the proposal even more similar to that of Bianchi et al).

3bis) One other important point is that earlier works such as Korba and Portier or Dai et al. promote a choice of $\epsilon $ that changes during the course of the algorithm (either $\epsilon $ goes to $1$ or $0$, respectively) while in the authors contribution, as well as in Bianchi et al, $\epsilon $ remains fixed. See also Chopin, N., Crucinio, F. R., & Korba, A. (2023). A connection between tempering and entropic mirror descent. arXiv preprint arXiv:2310.11914, where $\epsilon$ is updated during the algorithm. I think that the authors might clarify this point.

4) After those SMC or nonparametric adaptive importance sampling steps, the authors fit a parametric family. This way, the method is somewhat different than existing ones as we have a clear departure from nonparametric AIS or SMC-like approaches (in my opinion). For this reason, I encourage the authors to consider removing the variational family step from the algorithm—at least in the numerical experiments. Doing so would make it possible to better assess its impact. The algorithm I have in mind here is the same as their algorithm but with $\mu_{t+1} = F_{\mu_t} ^N$ plus some moves according to $L$ so as to allow some additional mutation (especially from the first component of the mixture) while keeping access to evaluation of the proposal. (Note that in this case, sampling $Z_t^i$ is no longer needed anymore.)

5) (following previous comments) Another point that seems to be missing concerns the reweighting step. After moving the particles using Langevin updates, a reweighting is applied, but no justification is provided. In standard importance sampling, reweighting is used to obtain unbiased (or nearly unbiased) estimates. Here, however, the target distribution is already involved in the particle updates through the gradient, which raises the question of whether this reweighting step is still necessary. Moreover, reweighting is not typically part of standard Langevin implementations. Some clarification or justification on this point would therefore be welcome.

6) The introduction of Langevin steps is motivated by the goal of improving exploration of the domain (as discussed in Section~2.3). The authors state that this is intended to allow ''non-local moves'' and ''global proposals''. but the connection with the proposed mechanism is not entirely evident. In particular, the example provided may not be the most appropriate. A well-known limitation of the Metropolis--Hastings algorithm is its tendency to become trapped in local modes. In the proposed algorithm, however, the Langevin updates correspond essentially to local moves, since the step size $\gamma$ is assumed to be small. As a result, it seems that the ability of the algorithm to eventually escape local minima may instead come from the use of the variational family, although this point is not explicitly discussed. Clarifying this aspect would strengthen the argument.

7) Their is one point in the numerical experiments that is I don't understand. Page 7 it is claimed that ``$\lambda = 1$ results in poor convergence'' and before it was claimed that $N_K$ kernel moves are performed. It seems quite unfair for the case $\lambda = 1$ as it doesn't use any of those kernel moves (which are not priceless computationally). For this reason, I believe that the algorithm with $\lambda = 1$ should be given some more steps.

8) In most competitors, except NUTS, no gradient is used. This looks like different settings in term of available information. Moreover, if I understand correctly, no variational family is used when running AIS. I found this quite unfair as AIS gives new particles. Finally, if my comment above is valid, I would suggest to include another competitor: $\lambda = 0$ with $n_K$-times more particles so the number of eval of $f$ or gradient would match.

9) In Theorem 3.3 isn't disappointing that the bound involves the KL between $\pi_\gamma$ and $\mu_0^R$. One would expect a KL between $\pi $ and $\mu_0^R$. Isn't there any bound known concerning the distance between $\pi_\gamma$ and $\pi$?

10) page 5 line 271 right column. There is a ``." that should be removed

---

> ### Author Rebuttal · Authors · 2026-03-30
>
> We thank the reviewer for the very careful reading. We greatly appreciate the positive assessment of the theoretical contribution, and the detailed and constructive comments.
>
> 0. The existence results are constructive only at the proof level: admissible $\lambda_t$ given in proofs depend on intractable expectations (e.g.,$\int f_t^\epsilon{\rm d}(\mu_tK_\pi)$).  In practice, $\lambda_t$ is a tuning parameter balancing exploitation (mirror step) and exploration (kernel step). If too large, exploration is weak; if too small, contraction deteriorates. We believe our result is important since it establishes a range of $\lambda_t$ ensuring contraction. But optimal tuning of $\lambda_t$ (done jointly with other hyperparameters), is challenging and left for future work. We updated the limitations and discussions at the end of the paper to clarify this.
>
> 1. We agree EM2C uses $\nabla\log\pi$ but only with Langevin kernels. In Section 4.2, we compare EM2C using RW (no gradient) kernel to both gradient-based (NUTS) and gradient-free (RW, AIS) methods. Our goal was to show that, even with limited information, EM2C performs competitively. We clarified this in the revision.
>
> 2. Our theory studies an idealized contractive map (not implementable). Update (2) cannot be sampled from exactly in general. In practice, we use weighting and resampling steps. The role of ${\sf F}_\theta$ is not covered by the theory: it is introduced to obtain a tractable proposal $(\tilde\mu_t)$ that can be sampled from and evaluated. We agree that the current version does not sufficiently separate these two levels. In the revision, a complete paragraph clarifies (i) the theoretical results obtained for the "oracle" algorithm and (ii) this projection is a practical approximation layer the practical Monte Carlo and optimization steps used to provide an efficient algorithm. Although these steps are not covered by the theory several directions for future works are discussed to control these approximations. Extending the analysis to include projection error is a major direction for future work.
>
> 3. We thank the reviewer for the reference that is closely related to Korba and Portier (2022), with a refined update of the proposal distribution. We mentioned it in the introduction. A key difference with Bianchi et al. (2024), is the reweighting of the particules pertubated by the Markov kernel. The local move $L$ is optional and only improves numerical stability.
>
> 3bis. We agree that several works consider time-varying $\epsilon_t$. We focus on fixed $\epsilon$ to highlight contraction, but the analysis extends to varying $\epsilon_t$. We clarified this.
>
> 4. We thank the reviewer for this suggeston. However, if we understand it correctly, setting $\mu_{t+1}=\hat{\mathcal F}^N_{\mu_t}$ is not straightforward. While $X_{t+1}^{1:N}$ can be sampled by resampling, mutated particles $Y_{t+1}^{1:N}$ lie outside the support of the empirical measure with probability one (for continuous kernels). Hence $d\pi/d\mu_{t+1}$ is not finite and weights are not well defined at these points. Thus, the empirical proposal is not suitable for iterating the reweighting step. This motivates the projection onto a tractable family. A nonparametric variant with smoothing would be an interesting direction.
>
> 5. In the revision, we precised that the reweighting is not for unbiasedness as in importance sampling. Its role is to adapt the proposal. If the kernel (not necessarily Langevin) discovers regions where $\mu_t$ is too small but $\pi$ is relatively large, then $d\pi/d\mu_t$ is large there and those particles should receive more mass. Thus, the kernel explores, while the weights decide which discovered regions should matter for the next proposal. We clarified this.
>
> 6. We agree Langevin steps are local. In EM2C, non-local behavior arises from reweighting: local moves generate candidates, and weights amplify particles reaching underexplored regions. Even rare particles can significantly affect the next proposal. The projection stabilizes this but does not discover modes alone.
>
> 7-8. To match computational budget (number of likelihood or gradient evaluations) between baseline, we increased the number of iterations for $\lambda=1$. Despite this, the method still performs worse (hence the importance of the Markov perturbation kernel). We plan to rerun AIS under a matched computational budget.
>
> **Additional remark :** Following other reviewers’ suggestions, we added a diffusion-based baseline (DiGS, Chen et al., 2024). See reviewer JC2J for details
>
> 9. The contraction is toward $\pi_\gamma$, not toward $\pi$. However the missing link is precisely Prop A.1, which bounds ${\rm KL}(\pi_\gamma\|\pi)$, i.e., the ULA bias (2nd term in the bound of Thm 3.3). We state this more explicitly.
>
> 10. Thank you. We corrected the typo.
>
> We believe the clarifications and additions described above, addresses the reviewer’s concerns and further strengthens the paper. We can discuss further some points.

---

> > ### Author Rebuttal · Reviewer_BwJ5 · 2026-04-01
> >
> > My concerns have been addressed by the authors rebuttal. As a consequence, I plan to increase my grade by 1 points.
> >
> > In my point 4: I was not clear. I had in mind a nonparametric version by replacing the parametric update by a kernel smoothing estimator. I am not sure exactly how to make it right within the authors algorithm. In fact, I still feel that the parametric update is playing a leading role in the proposal and it is somehow orthogonal to the SMC-MCMC moves that are conducted.

---

> > > ### Author Response · Authors · 2026-04-03
> > >
> > > We thank the reviewer for the positive and encouraging follow-up.
> > >
> > > We appreciate the clarification on point 4. A variant based on kernel smoothing is indeed an interesting direction. In our current implementation, the parametric update plays a central role as it is paramount to compute the weight in the subsequent iteration. We agree that exploring non parametric alternatives and understanding how they could be integrated within the proposed scheme is interesting to further address the trade-off between flexibility and tractable density evaluation, which is non trivial in the present context.

---

### Official Review · Reviewer_JC2J · 2026-03-10

**Soundness:** 3
**Presentation:** 2
**Significance:** 2
**Originality:** 3
**Overall Recommendation:** 3
**Confidence:** 3

**Summary:**

This paper introduces Entropic Mirror Monte Carlo (EM2C), an adaptive importance sampling method designed to sample from complex, multimodal distributions. The algorithm constructs a sequence of proposal distributions by combining Entropic Mirror Descent reweighting for exploitation with Markov kernels (like Langevin dynamics) for exploration. The authors prove that this idealized sequence converges geometrically fast to the target distribution under mild assumptions. Practically, EM2C demonstrates superior mode recovery and convergence compared to baselines on various benchmarks.

**Compliance With Llm Reviewing Policy:**

Affirmed.

**Final Justification:**

I acknowledge the additional Bayesian neural network result. However, I must note that the rebuttal engages in question reframing rather than direct response. The rebuttal answered a different question. My initial review explicitly requested validation on "real-world datasets" (plural) at high dimensionality ($d \sim 20{,}000$). The authors responded by asserting that "controlled multimodal experiments are sufficient" and that "multimodality and dimensionality are different challenges.


The claim that "multimodality and dimensionality are separate challenges" justifies testing only the former at $d=200$, but this ignores that high-dimensional multimodality (where gradients become uninformative) is precisely where the challenging problem lies. For a venue where Scalable Bayesian deep learning and molecular design are active frontiers, a sampling method validated only on synthetic Gaussian mixtures and a single moderate BNN falls short of the empirical bar. Solving low-dimensional multimodality is already handled by annealing and Parallel Tempering; the gap is scalability. If EM2C is truly a method for "sampling from complex, multimodal distributions in high-dimensional spaces" (the abstract), it must be validated on problems that are both multimodal and high-dimensional. I don't buy the "different tools" defense.


The rebuttal cites a few works to claim "moderate-dimensional benchmarks are standard." This in my opinion is selectively interpretative. Recent ICLR/NeurIPS/AISTATS sampling papers, operate at significantly higher dimensions and/or on real data[1,2,3].


That said, I will keep my score. I encourage the authors to pursue broader real-world validation for a future venue.

 [1] Metropolis Adjusted Microcanonical Hamiltonian Monte Carlo, NeurIPS 2025( d=2429)

 [2] Faster parallel MCMC: Metropolis adjustment is best served warm, AISTATS 2026(d=2519)

 [3] ENTROPY-MCMC: SAMPLING FROM FLAT BASINS WITH EASE, ICLR 2024 (d≈2.5×10^7)

**Key Questions For Authors:**

1. Since the paper lacks experiments on real-world datasets, it would be nice to include sampling examples for high dimensional examples( like d~20,000 in bayesian neural networks say) as a stress-test.

2. how does EM2C compare against more recent methods designed for mode identifiability such as Diffusive Gibbs Sampling (DiGS), SG-MCMC[2] ?

3. What is $\eta_k$ on Page 5?

[2] Cyclical Stochastic Gradient MCMC for Bayesian Deep Learning, Ruqi Zhang et. al, ICLR 2020.

**Limitations:**

Yes.

**Strengths And Weaknesses:**

Soundness: The submission is technically sound, with a clear distinction between its theoretical strengths and experimental limitation. The theoretical framework is undoubtedly the strongest aspect of this work. EM2C's  mixing mechanism  as discussed in Equation (3), is mathematically sophisticated and intuitive.  The proofs appear correct, relying on standard and valid assumptions that are typical for establishing geometric convergence in this domain( Theoerm 3.3).  The authors demonstrate a deep understanding of probability theory, and the theoretical claims are rigorous.

 While the experimental design is methodologically consistent (like using appropriate metrics like Sliced Wasserstein distance), the setups themselves are heavily contrived. The reliance on low-dimensional, syntehtic( d=2,10,20) Gaussian mixtures makes the empirical results feel less convincing. Furthermore, the baselines are somewhat limiting; the paper compares primarily against standard MCMC and AIS, missing comparisons to more modern,  multimodal samplers (e.g., methods similar to DIGS[1]) that would provide a stronger benchmark.

 Presentation: The submission is generally well-written and structured, with a particularly high standard of clarity in the theoretical sections namely sections 2 and 3. The mathematical exposition is excellent, with rigorous derivations and attention to detail. The appendix is comprehensive, providing detailed hyperparameter settings, kernel choices, and initialization strategies for all experiments( Tables 3,4,5).

As a suggestion, the 'problem-setup' can be expanded: there is very limited discussion on the shortcomings of sampling from multimodal distributions( Page 1, Right Column, 3 para.) before delving into EM2C. Another thing is, the core problem of multimodality and the specific limitations of standard Entropic Mirror Descent (which necessitate the proposed strategy) are discussed quite late in Section 2.3. I would suggest building the intuition first, and then suggesting how EM2C comes to rescue. One way to do this is by introducing Figure 1( or another motivational figure) early on, which provides a crucial visual intuition for the algorithm. This does not appear until Page 3. Also, explicitly italicizing or highlighting the core problem statement or the specific "gap" in current literature earlier in the introduction, right before introducing EM2C, would make the paper more accessible and palatable to audience beyond those in sampling theory.

Significance: From a theoretical standpoint, the paper does unlock an exciting direction by rigorously bridging optimization (Entropic Mirror Descent) and sampling (MCMC). The idea of constructing a clever geometrically converging sequence of densities via a mirror map is mathematically sophisticated compared to heuristic modifications of MCMC.

However, the challenge of sampling from multimodal distributions, is especially pronounced in high-dimensional settings like Bayesian deep learning and molecule generation. As the authors themselves note in the discussion( section 5), applying this to deep architectures is left for future work, which significantly reduces the practical significance for the broader machine learning community.

Originality: While the algorithmic toolkit(mirror descent, importance sampling, Langevin MCMC) is standard, the specific construction of the update rule combining a Markov kernel exploration step with a mirror descent reweighting step is a pleasant and theoretically principled approach.  EM2C aims to achieve the best of both worlds i.e. geometric convergence from mirror descent and robust mode discovery from MCMC. In this light, this work does provide new insights and does deepen understanding. It contextualizes itself, against related literature, like Annealed Importance Sampling (AIS), but this still can be improved( please see my questions below ).

[1] Diffusive Gibbs Sampling. Wenlin Chen et. al, ICML 2024.

---

> ### Author Rebuttal · Authors · 2026-03-30
>
> We thank the reviewer for the careful reading, the strong positive assessment of the theory and for the constructive comments on experiments and presentation. We have taken them into account to strengthen the paper.
>
> **Answer to the comments.**
>
> We added several important remarks in the revision of the paper to include reviewers comments. To follow comments on presentation and positioning, we expanded the discussion in the introduction and especially explain why standard approaches (mirror descent alone or MCMC alone) are insufficient to deal with multimodality. The reference to Figure 1 is now used from the introduction to provide a better intuition or motivation before formal developments. We point out more explicitly the core gap between existing methods that either rely on variational solution or MCMC methods, but not both simultaneously in a principled way.
>
> **Answer to the questions.**
>
> 1. The reviewer notes that our paper does unlock an exciting direction by rigorously bridging optimization and sampling. The  theoretical foundation of the proposed approach is indeed our main contribution. We agree that the current empirical section focuses on controlled synthetic benchmarks, and that extending evaluation to more challenging settings would strengthen the paper. We would also like to stress that multimodality is a challenge even for moderate dimensions and is still considered as standard settings in recent works, see for instance [1] for dimension ranging from 4 to 64. In addition, [2] also considered a 2-Dimensional mixture of Gaussian distributions with 40 mixture components.
>
> For our choice of Gaussian mixtures (in $d = 2,10,20$), although the ambient dimension is moderate, the number of modes grows exponentially with dimension. This provides an extremely challenging multimodal settings, where the difficulty stems from combinatorial mode structure rather than ambient dimension alone. In order to address the reviewer concern about scaling to very high-dimensional, we have added a benchmark in dimension $d=200$, used in Wenlin Chen et. al, ICML 2024.
>
> 2. We agree it would strengthen the empirical comparison to have additional baselines. Thank you for pointing Wenlin Chen et. al, ICML 2024. We added this diffusion-based Markov chain Monte Carlo (DiGS) baseline in the revised version. This approach demonstrates strong performance, successfully capturing the multimodal structure of the target while maintaining good sample diversity. DiGS significantly improves over AIS and other considered baselines, but is still outperformed by our algorithm in terms of discrepancy (SW$_2$, ED) across all settings. Results for two rings and two moons is presented below. We also increased the dimensionality of the mixture of Gaussian setting by considering a case with $d=200$ supporting the performance evaluation highlighted in the paper. For the revision, we have not consider SG-MCMC methods as they are primarily designed for large-scale settings with stochastic gradients, while our current setup assumes access to $\nabla \log \pi$.
>
> ### Dual Moons
>
> | Metric | RW   | NUTS | AIS  | DiGS  | EM2C |
> |--------|------|------|------|-------|------|
> | SW_2   | 0.62 | 1.74 | 0.17 | 0.074 | **0.071** |
> | ED     | 0.082| 1.48 |0.005 |0.0005 | **0.0004** |
>
> ### Two Rings
>
> | Metric | RW   | NUTS | AIS  | DiGS  | EM2C |
> |--------|------|------|------|-------|------|
> | SW_2   | 0.55 | 1.42 | 0.65 | 0.053 | **0.049** |
> | ED     | 0.056|0.441 |0.141 |0.0008 | **0.0006** |
>
> 3. We thank the reviewer for pointing out that the notation $\eta_k$ is not defined. It refers to the parameter of the base distribution. We now define it at first occurrence in the revised version.
>
> **Additional remark.**
>
> Finally, following additional reviewer suggestions, we improved the fairness of the experimental comparisons (matched number of likelihood or gradient evaluations). For instance, for $\lambda = 1$, we increased the number of iterations to better account for the unused kernel steps. We refer to our response to Reviewer BwJ5 for further details.
>
> We believe the clarifications and additions described above, addresses the reviewer’s concerns and further strengthens the paper.
>
> **References**
>
> [1] Improving the evaluation of samplers on multi-modal targets, Grenoux et al., ICLR 2025.
>
> [2] Diffusive Gibbs Sampling, Chen et al., ICML 2024.

---

> > ### Author Rebuttal · Reviewer_JC2J · 2026-04-02
> >
> > Thank You authors for responding to my rebuttal.
> >
> > I am glad you found my suggestions useful to strengthen the manuscript.
> >
> > However, my core concerns regarding empirical validation remain unsolved.
> >
> > 1. Empirical Validation and Scalability: While I acknowledge the addition of the DiGS baseline and the $d=200$ experiment, I must respectfully disagree with the framing that increasing modes keeping dimensionality fixed substitutes for challenging benchmarking. In high-dimensional settings ($d \sim 20,000$ as in Bayesian neural networks), the fundamental difficulty shifts: gradients become less informative about which direction to explore, and the geometry of the target becomes significantly more complex.
> >
> > As far as comparison with DiGS is concerned,  Chen et al. (2024) also demonstrate their method on real-world discrete structures and high-dimensional applications($d=550$). The absence of any real-world dataset limits the practical significance for the broader ML community, which was my primary concern in the initial review. Ultimately, I do not think I really got what I asked for, both in terms of dimensionality(20,000 vs 200) and real-world data use.
> >
> > 2. Theoretical Contribution: I have appreciated the paper on its theory and interesting combination. However, I would agree with my fellow Reviewer BwJ5 that the theory is mathematically sound but routine and not conceptually elevating('fairly elementary').  For a paper where empirical validation is limited to synthetic benchmarks, I would expect the theoretical contribution to either demonstrate novel proof techniques, sharp novel bounds that significantly advance our mathematical understanding. The current analysis, while rigorous, does not reach that threshold for me.
> >
> > The paper presents a theoretically principled approach with clear mathematical exposition, and I commend the authors for that. However, for ICML, a venue where practical impact and scalability are increasingly important—a sampling method tested only on synthetic problems up to $d=200$ without real-world validation presents a significant limitation. I encourage the authors to pursue this promising direction with more comprehensive empirical evaluation in future work. For the time being, I maintain my original assessment.

---

> > > ### Author Response · Authors · 2026-04-03
> > >
> > > We thank the reviewer for the follow-up and for reiterating the importance of empirical validation and scalability. We fully agree that these aspects are important and can further strengthen the paper, and we appreciate the opportunity to clarify our positioning.
> > >
> > > **On empirical validation and dimensionality**
> > >
> > > We took the reviewer’s initial concerns seriously and strengthened the empirical part by adding a DiGS baseline and increasing the dimensionality of our experiments (up to d=200).
> > >
> > > Following the reviewer’s suggestion, we evaluated EM2C on the Bayesian neural network experiment of DiGS (d = 550), using the same benchmark and comparable protocol (in particular identical MALA kernel settings as in DiGS).
> > > EM2C (300 iterations, $N=500$ particles) achieves a lower NLL than DiGS while requiring about $2\times$ fewer target energy evaluations, accounting for the projection step (mixture of 20 Gaussians). This shows that EM2C remains competitive in higher dimensional inference settings while being more computationally efficient. These results will be included in the revised version. A preliminary summary is provided below:
> > >
> > > **BNN (d = 550)**
> > >
> > > | Sampler | NLL |
> > > |--------|------|
> > > | DiGS   | 0.1886 ± 0.002 |
> > > | **EM2C** | **0.1509 ± 0.013** |
> > >
> > > While we addressed the reviewer’s request by adding a high-dimensional Bayesian neural network benchmark (d = 550), we respectfully disagree with the implication that controlled multimodal experiments are insufficient to assess sampling methods.
> > > In the Monte Carlo and sampling literature, including recent works at ICML, NeurIPS, ICLR and AISTATS, it is common practice, especially for theoretically grounded methods, to evaluate performance on controlled yet challenging benchmarks. High-dimensional real-world tasks and multimodal settings correspond to different difficulties and are typically studied with different methodological tools.
> > > Importantly, multimodality is widely recognized as a fundamental challenge that does not reduce to dimensionality. Recent works explicitly separate these aspects. For instance, [1] distinguishes mode structure and dimension and evaluates multimodal targets up to $d=256$. Similarly, [2,3] consider Gaussian mixtures up to $d=64$, and [4] studies Gaussian mixtures in $d=16$ and Bayesian logistic regression in $d=22$.
> > > We therefore believe that controlled multimodal benchmarks remain essential, and that our empirical validation, now complemented with a higher dimensional Bayesian inference benchmark, is aligned with current practice in the field.
> > >
> > > **On the theoretical contribution**
> > >
> > > We respectfully disagree with the characterization of the theory as “routine” or “elementary.” While the individual ingredients are classical, the core contribution lies in their principled and nontrivial combination, leading to a new iterative scheme in a relatively unexplored setting, and enjoys provable geometric convergence guarantees.
> > >
> > > To the best of our knowledge, establishing such guarantees for a method that explicitly combines mirror descent reweighting with MCMC exploration is not standard. The analysis requires controlling the interaction between optimization dynamics and stochastic transitions, which introduces nontrivial dependencies absent in either framework alone.
> > >
> > > We agree that the proofs rely on standard assumptions. While some technical aspects may appear standard in hindsight, their combination within our proposed framework is both meaningful and non-obvious. We believe the contribution lies in the structure of the framework and the guarantees it enables, rather than in introducing entirely new proof techniques. Many impactful works in machine learning theory follow a similar pattern, where new algorithmic insights are supported by rigorous, though not necessarily novel, analytical tools, and are valued for the perspectives and directions they open.
> > >
> > > Finally, we emphasize that our results pave the way for further research, in particular toward the analysis of the fully adaptive Monte Carlo implementation (Alg. 1), which remains a challenging and largely open problem that could be of interest to the ML community.
> > >
> > > We hope this clarification better conveys the intent and contribution of our work, and that the paper may benefit from a revised assessment in light of these points.
> > >
> > > [1] Improving the evaluation of samplers on multi-modal targets, Grenioux et al. (ICLR 2025)
> > >
> > > [2] Learned Reference-based Diffusion Sampling for Multi-modal Distributions , Noble et al. (ICLR 2025)
> > >
> > > [3] Sampling from Multi-modal Distributions via Reverse Diffusion, Vacher et al. (NeurIPS 2025)
> > >
> > > [4] Adaptive Importance Sampling meets Mirror Descent: a Bias-variance trade of, Korba and Portier (AISTAT 2022)

---

### Official Review · Reviewer_ab5E · 2026-03-13

**Soundness:** 2
**Presentation:** 3
**Significance:** 2
**Originality:** 2
**Overall Recommendation:** 4
**Confidence:** 3

**Summary:**

The paper introduces the development of a scheme to adaptively determine suitable proposal distributions for importance sampling based estimators of functionals. This has previously been explored as adaptive importance sampling in the literature, wherein the proposal distribution is adapted based on samples observed over iterations. In this work, the authors construct a sequence of proposal distributions based on a Markov chain. However, the contributions of the paper slowly morph into the development of a sampling procedure from the target distribution if I'm not mistaken.

**Compliance With Llm Reviewing Policy:**

Affirmed.

**Final Justification:**

I believe that the paper is interesting, and after reading the authors responses, I feel that a more detailed discussion of (1) computational costs is necessary, and (2) choosing hyperparameters. For (2), the authors seem to have a heuristic, but in the revised version. For (1), this is not discussed sufficiently in detail since the projection step adds a layer of complexity, which is not necessarily captured by number of gradient evaluations alone (in particular for the normalizing flows). One way to achieve this is by showing wall-clock times for the algorithms involved, or by discussing the complexity of EM2C as the sum of complexities of the key steps of the method.

Given the effort the authors have put into the rebuttal, I'm inclined to increase my score my a single point as this reflects my confidence in the authors fulfilling (1) and (2).

**Key Questions For Authors:**

1. Is there a specific reason why we care about $\mathsf{KL}(\pi || .)$ as opposed to $\mathsf{KL}(. || \pi)$? The latter is more commonly studied in the theoretical MCMC community, and there is a sound understand of a variety of MCMC methods in this setting.

2. Why do we care about a sequence $\{\mu_{t}\}$ that produces geometrically decaying discrepancy to $\pi$ (or in other words, a contracting map $\mathcal{F}$? Wouldn't a $1/poly(t)$ rate be also a suitable goal?

3. Can the authors provide more insight into how $\mathcal{F}_{e}$ can be implemented using Monte Carlo methods? Does that not assume access to a density of $\mu$ as well, which may not be available even after a single iteration for most $\pi$?

4. How tractable is it to find a sequence $\{\lambda_{t}\}$ such that the mapping $\mathcal{F}_{em}$ is contractive?

5. Is there an interpretation of $\mathcal{F}_{em}$ as performing entropic mirror descent w.r.t. another functional like the KL, or is it mirror descent w.r.t. a different mirror function than the entropy?

6. Are there any assumptions on the step size $\gamma$ for Theorem 3.3?

**Strengths And Weaknesses:**

The paper is very well written, and is rather clear in its exposition, which is highly appreciated! The key mathematical quantities are defined suitably, and the central goal of the paper is described clearly. The basis of the innovation of the paper which is in how the idealized entropic mirror descent mapping produces a contractive mapping $\mathcal{F}$ is stated in an intuitive manner (although it would help if a derivation of (2) corresponding to a mirror descent step is stated more explicitly, even if referencing a theorem / proposition from Korba and Portier). One thing that is not particularly evident to me is if (3) has to be considered as a mixture of two distributions or a weighted average of two densities?

One thing that is unclear to me is how Theorem 3.3 (the main theoretical contribution of this work) differs from mixing time results for the unadjusted Langevin algorithm (see for instance Vempala and Wibisono (2023)), or rather why the sequence $\{\mu_{t}\}$ generated by the recursion above Theorem 3.3 is more useful that the sequence of distributions generated by the unadjusted Langevin algorithm directly. It is well known from the guarantees of the unadjusted Langevin algorithm (Theorem 1, Vempala and Wibisono (2023)) that the convergence to the bias limit is exponentially quick, which is also observed in the statement of Theorem 3.3.

The one major qualm I have about the paper is that the focus of the paper shifts from estimating functionals to performing sampling as highlighted in the summary. How does this scheme enables better estimation of functionals, or how are the distributions generated used in importance sampling. I find that neither of these directions are discussed past Section 1.

In the empirical section of the paper, the authors describe the empirical setups well and discuss parameter choices explicitly.

Minor nits:
1. The unadjusted Langevin algorithm is attributed to Roberts and Tweedie (1996).

---

> ### Author Rebuttal · Authors · 2026-03-30
>
> We thank the reviewer for the positive comments on the writing and exposition, and for the insightful questions.
>
> **Answer to the comments.**
>
> Eq (2) can be made more explicit. The density is $\mu_{t+1}(x)\propto\pi(x)^\epsilon\mu_t(x)^{1-\epsilon}$. This is the entropic mirror-descent update for minimizing $\mu\mapsto{\rm KL}(\pi\Vert\mu)$. We added this derivation in the revision. In addition, Eq. (3) is a convex combination of measures; when densities exist, it is a weighted average of densities which is stated now clearly in the paper.
>
> Relation to ULA: we agree that if the goal were only asymptotic sampling, ULA is a natural baseline, but it may perform poorly in multimodal settings (see Figure 10). Our goal is different: to construct a sequence of proposal distributions for importance sampling. We define $(\mu_t)$ via a map combining entropic reweighting and a Markov step, and prove contraction for $\pi$-invariant kernels (Prop. 3.2) and ULA (Thm. 3.3). Our goal differs from classical ULA analyses, as we study a sequence of proposals produced by our map. The geometric decrease comes from the entropic mirror map, while the Markov kernel component is added so that it preserves this contraction for a suitable choice of $\lambda_t$. Thus the kernel is not the sole driver of convergence. It improves exploration while preserving mirror-descent structure. We now made this distinction clearer.
>
> Thank you for spotting this; we now cite Roberts and Tweedie (1996).
>
> **Answer to the questions.**
>
> 1. The focus on ${\rm KL}(\pi\Vert\mu)$ comes from mirror-descent and IS perspectives. IS is controlled by the mismatch from target to proposal. Controlling weights bias and variance relate to this KL. This is also the KL contracted by the entropic mirror map. We agree ${\rm KL}(\mu\Vert\pi)$ is standard in the MCMC literature, but it serves to study convergence of chain laws. We made this motivation more explicit.
>
> 2. The reviewer is right that a 1/poly(t) rate still implies asymptotic improvement and is meaningful in principle. However geometric contraction is valuable for quickly reducing weight degeneracy, especially in multimodal settings. Rapidly reducing the discrepancy between target and proposal helps the method to move out of a severe weight degeneracy regime. Moreover, the entropic mirror map yields a geometric contraction property, which we preserve in the proposed construction. We clarified that geometric decay is a particularly strong property available here.
>
> 3. The reviewer is right. The practical implementation requires pointwise evaluation of the current proposal $\tilde\mu_t$ to compute the weights. For a completely implicit proposal family with no tractable density, Algorithm 1 would not be implementable. This is precisely why the projection step is a major part of the algorithm: after resampling from the empirical mixture, we fit a tractable parametric surrogate whose density can be evaluated at the next iteration. This is now stated more explicitly in the revised version.
>
> 4. Regarding $\lambda_t$, the existence results are constructive only at the proof level: admissible $\lambda_t$ depend on intractable expectations (e.g., $\int f_t^\varepsilon\mathrm d(\mu_t K_\pi)$, see proof).  In practice, $\lambda_t$ is a tuning parameter balancing exploitation (mirror step) and exploration (kernel step), as in many adaptive Monte Carlo methods where theory ensures existence of a good schedule but implementation relies on heuristics. We believe that our result is important since it establishes that we can find a range of $\lambda_t$ that ensures contraction. However, optimal tuning of $\lambda_t$, that should be done jointly with the other hyperparameters, is challenging and left for future work. We updated the limitations and discussions at the end of the paper to clarify this.
>
> 5. The mapping $\mathcal{F}_{\rm em}$ should be viewed as a perturbation of the standard entropic mirror descent update rather than as mirror descent for a different functional. The reweighting by $(\pi/\mu)^\varepsilon$ preserves the entropic mirror descent step, which drives the KL contraction. The additional Markov kernel introduces a transport step that improves exploration but breaks the exact mirror descent interpretation.
>
> 6. Yes. The condition on $\gamma$ is given in Proposition A.1 but is indeed missing from Theorem 3.3. Thank for pointing this. We added the condition in the revision.
>
> **Additional remark.**
>
> Our mehtod outperforms some recent methods in challenging settings still commonly used in recent works even if the dimension remains small. However, to follow other reviewer's suggestions, we strengthened the empirical section by including a recent diffusion-based baseline (DiGS, Chen et al., 2024), and added an experimental setting in higher dimension ($d=200$). See reviewer JC2J.
>
> We believe the clarifications and additions described above, addresses the reviewer’s concerns and further strengthens the paper.

---

> > ### Author Rebuttal · Reviewer_ab5E · 2026-04-02
> >
> > Thank you for the rebuttal to my review; my question have been sufficiently addressed.
> >
> > I have a couple of reasons for why I feel that this paper would need a little more work.
> > 1. A discussion of computational costs: since there is a projection step involved, how does the computational cost of EM2C compare with the other baseline in this paper.
> > 2. Concrete suggestions for $\lambda_{t}$: since this is an important parameter, it would be useful to provide general guidance on how to pick this, or develop perhaps a semi-automatic tuning scheme for $\lambda_{t}$
> > 3. An ablation study for parametric families: the Gaussian Mixture family fundamentally depends on the number of components K. How does this affect the quality of the algorithm? Is there a general guidance here for the parameter that would be useful for a potential user of this method?
> >
> > I believe that addressing this would help strengthen the paper further; I do think that the authors perform a very good job in writing!

---

> > > ### Author Response · Authors · 2026-04-03
> > >
> > > We thank the reviewer for the positive feedback and for the constructive suggestions on how to further strengthen the paper.
> > >
> > > **Computational cost**
> > >
> > > We thank the reviewer for this additional suggestions, which help further clarify the practical aspects of the method. We agree that discussing computational cost is important. In the revised version, we ensured fair comparisons by matching the number of likelihood/gradient evaluations across methods. Regarding the projection step, its cost depends on the chosen parametric family. For Gaussian mixtures, it corresponds to fitting a standard density estimator (e.g., EM), while for normalizing flows it amounts to a training phase which is highly parallelizable and amortizable across iterations. Overall, this type of projection is comparable to standard density fitting steps used in adaptive importance sampling or variational inference. We took this into account in our experiments by allocating a slightly larger computational budget to methods that do not include a projection step, in order to ensure a fairer comparison.
> > >
> > > **Tuning $\lambda_t$**
> > >
> > > We agree that providing guidance is important. The parameter $\lambda_t$ controls the trade-off between reweighting and exploration. Beyond this qualitative interpretation, the proof provides useful insight: $\lambda_t$ requires fine tuning only when the expectation under $\mu_t K_\pi$ of the Radon-Nikodym derivative of $(d\pi/d\mu_t)^\epsilon$ exceeds 1. Otherwise, the geometric contraction occurs for any $\lambda_t \in (0, 1]$. We pointed that out more explicitly in the revision.
> > >
> > > In practice, we observe that simple choices already perform well. A useful heuristic is:
> > > (i) larger $\lambda_t$, stronger reweighting, faster concentration but risk of reduced exploration, (ii) smaller $\lambda_t$, more exploration via the kernel, but slower contraction.
> > > To find a trade-off, one can monitor weight degeneracy via effective sample size. This monitoring is provided in the revised version. A low effective sample size typically indicates insufficient exploration and suggests decreasing $\lambda_t$. However this is a simple heuristic. Designing fully automatic tuning for $\lambda_t$ is an interesting direction for future work, but would require substantial additional developments beyond the scope of this work.
> > >
> > > **Parametric family and choice of $K$**
> > >
> > > We would like to clarify that the method is not restricted to Gaussian mixtures. While GMMs are used as a simple and interpretable baseline, we also consider more expressive families such as normalizing flows, which do not require specifying the number of components $K$. In this sense, the projection step should be viewed as a generic density estimation component rather than a limitation of the method itself.
> > >
> > > In practice, we found that moderate overparameterization is preferable, as underparameterized models may fail to capture multimodal structure. This is consistent with standard practice in mixture modeling. We mentioned this point in the revision.
> > >
> > > We hope these additions better convey the intent and contribution of our work, and that the paper may benefit from a revised assessment in light of these points.

---

### Decision · Program_Chairs · 2026-04-30

**Decision:**

Accept (regular)

**Comment:**

This paper received mixed but overall positive recommendations after discussion. Reviewers agree that the paper is well written and that the theory for the idealized EM2C method is technically sound. The main remaining concern is the gap between the oracle-level theory and the practical algorithm. In particular, the Monte Carlo approximation and the projection step are not covered by the main theoretical guarantees.
After rebuttal, one reviewer moved to weak accept. Two reviewers indicated that the revisions addressed their concerns. One reviewer, however, maintained concerns about practical significance and empirical scope. On balance, I recommend acceptance as a theory-driven methods paper. For the final version, the authors should clearly state the scope of the theoretical guarantees. They should also explicitly acknowledge the remaining theory-practice gap. Claims about high-dimensional applicability should be moderated where appropriate. The paper should also incorporate the clarifications and additional empirical results provided during rebuttal.